# Identification and Estimation Of Causal Effects from Dependent Data

**Eli Sherman**
Department of Computer Science
Johns Hopkins University
Baltimore, MD 21218
esherman@jhu.edu

**Ilya Shpitser**
Department of Computer Science
Johns Hopkins University
Baltimore, MD 21218
ilyas@cs.jhu.edu

## Abstract

The assumption that data samples are independent and identically distributed (iid) is standard in many areas of statistics and machine learning. Nevertheless, in some settings, such as social networks, infectious disease modeling, and reasoning with spatial and temporal data, this assumption is false. An extensive literature exists on making causal inferences under the iid assumption [17, 11, 26, 21], even when unobserved confounding bias may be present. But, as pointed out in [19], causal inference in non-iid contexts is challenging due to the presence of both unobserved confounding *and* data dependence. In this paper we develop a general theory describing when causal inferences are possible in such scenarios. We use *segregated graphs* [20], a generalization of latent projection mixed graphs [28], to represent causal models of this type and provide a complete algorithm for non-parametric identification in these models. We then demonstrate how statistical inference may be performed on causal parameters identified by this algorithm. In particular, we consider cases where only a single sample is available for parts of the model due to *full interference*, i.e., all units are pathwise dependent and neighbors' treatments affect each others' outcomes [24]. We apply these techniques to a synthetic data set which considers users sharing fake news articles given the structure of their social network, user activity levels, and baseline demographics and socioeconomic covariates.

## 1 Introduction

The assumption of independent and identically distributed (iid) samples is ubiquitous in data analysis. In many research areas, however, this assumption simply does not hold. For instance, social media data often exhibits dependence due to homophily and contagion [19]. Similarly, in epidemiology, data exhibiting herd immunity is likely dependent across units. Likewise, signal processing and sequence learning often consider data that are spatially [8] or temporally [23] dependent.

In causal inference, dependence in data often manifests as *interference* wherein some units' treatments may causally affect other units' outcomes [3, 9]. Herd immunity is a canonical example of interference since other subjects' vaccination status causally affects the likelihood of a particular subject contracting a disease. Even under the iid assumption, making causal inferences from observed data is difficult due to the presence of unobserved confounding. This difficulty is worsened when interference is present, as described in detail in [19]. In general, these difficulties prevent identification of causal parameters of interest, making estimation of these parameters from data an ill-posed problem. An extensive literature on identification of causal parameters (under the iid assumption) has been developed. The *g-formula* [17] identifies any interventional distribution in directed acylcic graph-based (DAG) causal models without latent variables. Pearl showed that in certain cases identification is

possible even in the presence of unobserved confounding via the *front-door criterion* [11]. These results were generalized into a complete identification theory in hidden variable causal DAG models via the ID algorithm [26, 21]. An extensive theory of estimation of identified causal parameters has been developed. Some approaches are described in [17, 18], although this is far from an exhaustive list. While work on identification and estimation of causal parameters under interference exists [3, 25, 9, 14, 13, 7, 1], no general theory has been developed up to now. In this paper, we aim to provide this theory for a general class of causal models that permit interference.

## 2    A Motivating Example

To motivate subsequent developments, we introduce the following example application. Consider a large group of internet users, belonging to a set of online communities, perhaps based on shared hobbies or political views. For each user $i$, their time spent online $A_i$ is influenced by their observed vector of baseline factors $C_i$, and unobserved factors $U_i$. In addition, each user maintains a set of friendship ties with other users via an online social network. The user's activity level in the network, $M_i$, is potentially dependent on the user's friends' activities, meaning that for users $j$ and $k$, $M_j$ and $M_k$ are potentially dependent. The dependence between $M$ variables is modeled as a stable symmetric relationship that has reached an equilibrium state. Furthermore, activity level $M_i$ for user $i$ is influenced by observed factors $C_i$, time spent online $A_i$, and the time spent online $A_j$ of any unit $j$ who is a friend of $i$. Finally, we denote user $i$'s sharing behavior by $Y_i$. This behavior is influenced by the social network activity of the unit, and possibly the unit friends' time spent online.

A crucial assumption in our example is that for each user $i$, purchasing behavior $Y_i$ is causally influenced by baseline characteristics $C_i$, social network activity $M_i$, and unobserved characteristics $U_i$, but time spent online $A_i$ does not *directly* influence sharing $Y_i$, except as mediated by social network activity of the users. While this might seem like a rather strong assumption, it is more reasonable than standard "front-door" assumptions [12] in the literature, since we allow the entire social network structure to mediate the influence $A_i$ on $Y_i$ for every user.

We are interested in predicting how a counterfactual change in a set of users' time spent online influences their purchasing behavior. Note that solving this problem from observed data on users as we described is made challenging both by the fact that unobserved variables causally affect both community membership and sharing, creating spurious correlations, and because social network membership introduces dependence among users. In particular, for realistic social networks, every user's activity potentially depends on every other user's activity (even if indirectly). This implies that a part of the data for this problem may effectively consist of a single dependent sample [24].

In the remainder of the paper we formally describe how causal inference may be performed in examples like above, where both unobserved confounding and data dependence are present. In section 3 we review relevant terminology and notation, give factorizations defining graphical models, describe causal inference in models without hidden variables, and give identification theory for such models in terms of a modified factorization. We also introduce the dependent data setting we will consider. In section 4 we describe more general *nested* factorizations [16] applicable to marginals obtained from hidden variable DAG models, and describe identification theory in causal models with hidden variables in terms of a modified nested factorization. In section 5, we introduce causal chain graph models [6] as a way of modeling causal problems with interference and data dependence, and pose the identification problem for interventional distributions in such models. In section 6 we give a sound and complete identification algorithm for interventional distributions in a large class of causal chain graph models with hidden variables, which includes the above example, but also many others. We describe our experiments, which illustrate how identified functionals given by our algorithm may be estimated in practice, even in *full interference* settings where all units are mutually dependent, in section 7. Our concluding remarks are found in section 8.

## 3    Background on Causal Inference And Interference Problems

### 3.1    Graph Theory

We will consider causal models represented by mixed graphs containing directed ($\rightarrow$), bidirected ($\leftrightarrow$) and undirected ($-$) edges. Vertices in these graphs and their corresponding random variables will be used interchangeably, denoted by capital letters, e.g. $V$; values, or realizations, of vertices

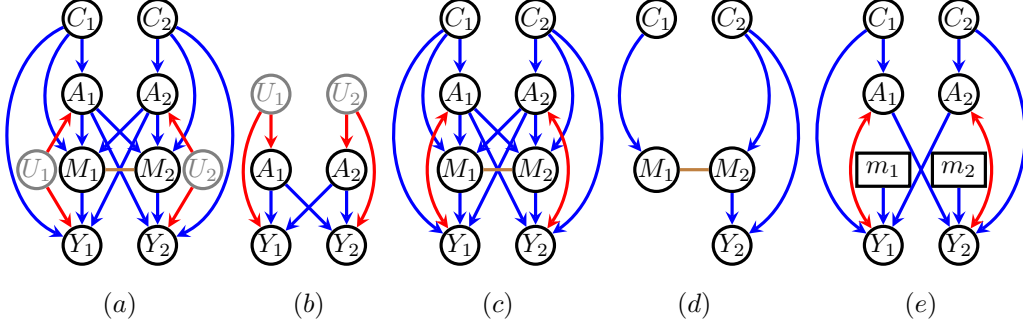

Figure 1: (a) A causal model representing the effect of community membership on article sharing, mediated by social network structure. (b) A causal model on dyads which is a variation of causal models of interference considered in [9]. (c) A latent projection of the CG in (a) onto observed variables. (d) The graph representing $\mathcal{G}_{\mathbf{Y}^*}$ for the intervention operation $\mathrm{do}(a_1)$ applied to (c). (e) The ADMG obtained by fixing $M_1, M_2$ in (c).

and variables will be denoted by lowercase letters, e.g. $v$; bold letters will denote sets of variables or values e.g. $\mathbf{V}$ or $\mathbf{v}$. We will denote the state space of a variable $V$ or a set of variables $\mathbf{V}$ as $\mathfrak{X}_V$, and $\mathfrak{X}_{\mathbf{V}}$. Unless stated otherwise, all graphs will be assumed to have a vertex set denoted by $\mathbf{V}$. For a mixed graph $\mathcal{G}$ of the above type, we denote the standard genealogic sets for a variable $V \in \mathbf{V}$ as follows: parents $\mathrm{pa}_{\mathcal{G}}(V) \equiv \{W \in \mathbf{V} | W \to V\}$, children $\mathrm{ch}_{\mathcal{G}}(V) \equiv \{W \in \mathbf{V} | V \to W\}$, siblings $\mathrm{sib}_{\mathcal{G}}(V) \equiv \{W \in \mathbf{V} | W \leftrightarrow V\}$, neighbors $\mathrm{nb}_{\mathcal{G}}(V) \equiv \{W \in \mathbf{V} | W - V\}$, ancestors $\mathrm{an}_{\mathcal{G}}(V) \equiv \{W \in \mathbf{V} | W \to \cdots \to V\}$, descendants $\mathrm{de}_{\mathcal{G}}(V) \equiv \{W \in \mathbf{V} | V \to \cdots \to W\}$, and non-descendants $\mathrm{nd}_{\mathcal{G}}(V) \equiv \mathbf{V} \setminus \mathrm{de}_{\mathcal{G}}(V)$. We define the *anterior* of $V$, or $\mathrm{ant}_{\mathcal{G}}(V)$, to be the set of all vertices with a partially directed path (a path containing only $\to$ and $-$ edges such that no $-$ edge can be oriented to induce a directed cycle) into $V$. These relations generalize disjunctively to sets, for instance for a set $\mathbf{S}$, $\mathrm{pa}_{\mathcal{G}}(\mathbf{S}) = \bigcup_{S \in \mathbf{S}} \mathrm{pa}_{\mathcal{G}}(S)$. We also define the set $\mathrm{pa}_{\mathcal{G}}^s(\mathbf{S})$ as $\mathrm{pa}_{\mathcal{G}}(\mathbf{S}) \setminus \mathbf{S}$. Given a graph $\mathcal{G}$ and a subset $\mathbf{S}$ of $\mathbf{V}$, define the *induced subgraph* $\mathcal{G}_{\mathbf{S}}$ to be a graph with a vertex set $\mathbf{S}$ and all edges in $\mathcal{G}$ between elements in $\mathbf{S}$.

Given a mixed graph $\mathcal{G}$, we define a *district* $\mathbf{D}$ to be a maximal set of vertices, where every vertex pair in $\mathcal{G}_{\mathbf{D}}$ is connected by a bidirected path (a path containing only $\leftrightarrow$ edges). Similarly we define a *block* $\mathbf{B}$ to be a maximal set of vertices, where every vertex pair in $\mathcal{G}_{\mathbf{B}}$ is connected by an undirected path (a path containing only $-$ edges). Any block of size at least 2 is called a non-trivial block. We define a *maximal clique* as a maximal set of vertices pairwise connected by undirected edges. The set of districts in $\mathcal{G}$ is denoted by $\mathcal{D}(\mathcal{G})$, the set of blocks is denoted by $\mathcal{B}(\mathcal{G})$, the set non-trivial blocks is denoted by $\mathcal{B}^{nt}(\mathcal{G})$, and the set of cliques is denoted by $\mathcal{C}(\mathcal{G})$. The district of $V$ is denoted by $\mathrm{dis}_{\mathcal{G}}(V)$. By convention, for any $V$, $\mathrm{dis}_{\mathcal{G}}(V) \cap \mathrm{de}_{\mathcal{G}}(V) \cap \mathrm{an}_{\mathcal{G}}(V) \cap \mathrm{ant}_{\mathcal{G}}(V) = \{V\}$.

A mixed graph is called *segregated (SG)* if it contains no partially directed cycles, and no vertex has both neighbors and siblings, Fig. 1 (c) is an example. In a SG $\mathcal{G}$, $\mathcal{D}(\mathcal{G})$ and $\mathcal{B}^{nt}(\mathcal{G})$ partition $\mathbf{V}$. A SG without bidirected edges is called a chain graph (CG) [5]. A SG without undirected edges is called an acyclic directed mixed graph (ADMG) [15]. A CG without undirected edges or an ADMG without bidirected edges is a directed acyclic graph (DAG) [10]. A CG without directed edges is called an undirected graph (UG). Given a CG $\mathcal{G}$, the augmented graph $\mathcal{G}^a$ is the UG where any adjacent vertices in $\mathcal{G}$ or any elements in $\mathrm{pa}_{\mathcal{G}}(\mathbf{B})$ for any $\mathbf{B} \in \mathcal{B}(\mathcal{G})$ are connected by an undirected edge.

## 3.2  Graphical Models

A graphical model is a set of distributions with conditional independences represented by structures in a graph. The following (standard) definitions appear in [5]. A DAG model, or a Bayesian network, is a set of distributions associated with a DAG $\mathcal{G}$ that can be written in terms of a DAG factorization: $p(\mathbf{V}) = \prod_{V \in \mathbf{V}} p(V | \mathrm{pa}_{\mathcal{G}}(V))$. A UG model, or a Markov random field, is a set of distributions associated with a UG $\mathcal{G}$ that can be written in terms of a UG factorization: $p(\mathbf{V}) = Z^{-1} \prod_{\mathbf{C} \in \mathcal{C}(\mathcal{G})} \psi_{\mathbf{C}}(\mathbf{C})$, where $Z$ is a normalizing constant. A CG model is a set of distributions associated with a CG $\mathcal{G}$ that

can be written in terms of the following two level factorization: $p(\mathbf{V}) = \prod_{\mathbf{B} \in \mathcal{B}(\mathcal{G})} p(\mathbf{B}|\operatorname{pa}_{\mathcal{G}}(\mathbf{B}))$, where for each $\mathbf{B} \in \mathcal{B}(\mathcal{G})$, $p(\mathbf{B}|\operatorname{pa}_{\mathcal{G}}(\mathbf{B})) = Z(\operatorname{pa}_{\mathcal{G}}(\mathbf{B}))^{-1} \prod_{\mathbf{C} \in \mathcal{C}((\mathcal{G}_{\mathbf{B} \cup \operatorname{pa}_{\mathcal{G}}(\mathbf{B})})^a); \mathbf{C} \not\subseteq \operatorname{pa}_{\mathcal{G}}(\mathbf{B})} \psi_{\mathbf{C}}(\mathbf{C})$.

## 3.3 Causal Inference and Causal Models

A causal model of a DAG is also a set of distributions, but on counterfactual random variables. Given $Y \in \mathbf{V}$ and $\mathbf{A} \subseteq \mathbf{V} \setminus \{Y\}$, a counterfactual variable, or 'potential outcome', written as $Y(\mathbf{a})$, represents the value of $Y$ in a hypothetical situation where a set of *treatments* $\mathbf{A}$ is set to values $\mathbf{a}$ by an *intervention operation* [12]. Given a set $\mathbf{Y}$, define $\mathbf{Y}(\mathbf{a}) \equiv \{\mathbf{Y}\}(\mathbf{a}) \equiv \{Y(\mathbf{a}) \mid Y \in \mathbf{Y}\}$. The distribution $p(\mathbf{Y}(\mathbf{a}))$ is sometimes written as $p(\mathbf{Y}|\operatorname{do}(\mathbf{a}))$ [12].

Causal models of a DAG $\mathcal{G}$ consist of distributions defined on counterfactual random variables of the form $V(\mathbf{a})$ where $\mathbf{a}$ are values of $\operatorname{pa}_{\mathcal{G}}(V)$. In this paper we assume Pearl's functional model for a DAG $\mathcal{G}$ with vertices $\mathbf{V}$, where $V(\mathbf{a})$ are determined by *structural equations* $f_V(\mathbf{a}, \epsilon_V)$, which remain invariant under any possible intervention on $\mathbf{a}$, with $\epsilon_V$ an exogenous disturbance variable which introduces randomness into $V$ even after all elements of $\operatorname{pa}_{\mathcal{G}}(V)$ are fixed. Under Pearl's model, the distribution $p(\{\epsilon_V | V \in \mathbf{V}\})$ is assumed to factorize as $\prod_{V \in \mathbf{V}} p(\epsilon_V)$. This implies that the sets of variables $\{\{V(\mathbf{a}_V) \mid \mathbf{a}_V \in \mathfrak{X}_{\operatorname{pa}_{\mathcal{G}}(V)}\} \mid V \in \mathbf{V}\}$ are mutually independent [12]. The *atomic counterfactuals* in the above set model the relationship between $\operatorname{pa}_{\mathcal{G}}(V)$, representing direct causes of $V$, and $V$ itself. From these, all other counterfactuals may be defined using recursive substitution. For any $\mathbf{A} \subseteq \mathbf{V} \setminus \{V\}$, $V(\mathbf{a}) \equiv V(\mathbf{a}_{\operatorname{pa}_{\mathcal{G}}(V) \cap \mathbf{A}}, \{\operatorname{pa}_{\mathcal{G}}(V) \setminus \mathbf{A}\}(\mathbf{a}))$. For example, in the DAG in Fig. 1 (b), $Y_1(a_1)$ is defined to be $Y_1(a_1, U_1, A_2(U_2))$. Counterfactual responses to interventions are often compared on the mean difference scale for two values $a, a'$, representing cases and controls: $\mathbb{E}[Y(a)] - \mathbb{E}[Y(a')]$. This quantity is known as the average causal effect (ACE).

A causal parameter is said to be *identified* in a causal model if it is a function of the observed data distribution $p(\mathbf{V})$. Otherwise the parameter is said to be *non-identified*. In any causal model of a DAG $\mathcal{G}$, all interventional distributions $p(\mathbf{V} \setminus \mathbf{A}|\operatorname{do}(\mathbf{a}))$ are identified by the *g-formula* [17]:

$$p(\mathbf{V} \setminus \mathbf{A}|\operatorname{do}(\mathbf{a})) = \prod_{V \in \mathbf{V} \setminus \mathbf{A}} p(V|\operatorname{pa}_{\mathcal{G}}(V))\big|_{\mathbf{A}=\mathbf{a}} \tag{1}$$

Note that the g-formula may be viewed as a modified (or *truncated*) DAG factorization, with terms corresponding to elements in $\mathbf{A}$ missing.

## 3.4 Modeling Dependent Data

So far, the causal and statistical models we have introduced assumed data generating process that produce independent samples. To capture examples of the sort we introduced in section 2, we must generalize these models. Suppose we analyze data with $M$ blocks with $N$ units each. It is not necessary to assume that blocks are equally sized for the kinds of problems we consider, but we make this assumption to simplify our notation. Denote the variable $Y$ for the $i$'th unit in block $j$ as $Y_i^j$. For each block $j$, let $\mathbf{Y}^j \equiv (Y_1^j, \ldots, Y_N^j)$, and let $\mathbf{Y} \equiv (\mathbf{Y}^1, \ldots, \mathbf{Y}^M)$. In some cases we will not be concerned with units' block memberships. In these cases we will accordingly omit the superscript and the subscript will index the unit with respect to all units in the network.

We are interested in counterfactual responses to interventions on $\mathbf{A}$, treatments on all units in all blocks. For any $\mathbf{a} \in \mathfrak{X}_{\mathbf{A}}$, define $Y_i^j(\mathbf{a})$ to be the potential response of unit $i$ in block $j$ to a hypothetical treatment assignment of $\mathbf{a}$ to $\mathbf{A}$. We define $\mathbf{Y}^j(\mathbf{a})$ and $\mathbf{Y}(\mathbf{a})$ in the natural way as vectors of responses, given a hypothetical treatment assignment to $\mathbf{a}$, either for units in block $j$ or for all units, respectively. Let $\mathbf{a}^{(j)}$ be a vector of values of $\mathbf{A}$, where values assigned to units in block $j$ are *free variables*, and other values are *bound variables*. Furthermore, for any $\tilde{\mathbf{a}}^j \in \mathfrak{X}_{\mathbf{A}^j}$, let $\mathbf{a}^{(j)}[\tilde{\mathbf{a}}^j]$ be a vector of values which agrees on all bound values with $\mathbf{a}^{(j)}$, but which assigns $\tilde{\mathbf{a}}^j$ to all units in block $j$ (e.g. which binds free variables in $\mathbf{a}^{(j)}$ to $\tilde{\mathbf{a}}^j$).

A commonly made assumption is *interblock non-interference*, also known as *partial interference* in [22, 25], where for any block $j$, treatments assigned to units in a block other than $j$ do not affect the responses of any unit in block $j$. Formally, this is stated as $(\forall j, \mathbf{a}^{(j)}, \mathbf{a}'^{(j)}, \tilde{\mathbf{a}}^j), \mathbf{Y}^j(\mathbf{a}^{(j)}[\tilde{\mathbf{a}}^j]) = \mathbf{Y}^j(\mathbf{a}'^{(j)}[\tilde{\mathbf{a}}^j])$. Counterfactuals under this assumption are written in a way that emphasizes they only depend on treatments assigned within that block. That is, for any $\mathbf{a}^{(j)}$, $\mathbf{Y}^j(\mathbf{a}^{(j)}[\tilde{\mathbf{a}}^j]) \equiv \mathbf{Y}^j(\tilde{\mathbf{a}}^j)$.

In this paper we largely follow the convention of [9], where variables corresponding to distinct units within a block are shown as distinct vertices in a graph. As an example, Fig. 1 (b) represents a causal model with observed data on multiple realizations of *dyads* or blocks of two dependent units [4]. Note that the arrow from $A_2$ to $Y_1$ in this model indicates that the treatment of unit 2 in a block influences the outcome of unit 1, and similarly for treatment of unit 1 and outcome of unit 2. In this model, a variation of models considered in [9], the interventional distributions $p(Y_2|\text{do}(a_1)) = p(Y_2|a_1)$ and $p(Y_1|\text{do}(a_2)) = p(Y_1|a_2)$ even if $U_1, U_2$ are unobserved.

# 4 Causal Inference with Hidden Variables

If a causal model contains hidden variables, only data on the observed marginal distribution is available. In this case, not every interventional distribution is identified, and identification theory becomes more complex. However, just as identified interventional distributions were expressible as a truncated DAG factorization via the g-formula (1) in fully observed causal models, identified interventional distributions are expressible as a truncated *nested* factorization [16] of a *latent projection* ADMG [28] that represents a class of hidden variable DAGs that share identification theory. In this section we define latent projection ADMGs, introduce the nested factorization with respect to an ADMG in terms of a fixing operator, and re-express the ID algorithm [27, 21] as a truncated nested factorization.

## 4.1 Latent Projection ADMGs

Given a DAG $\mathcal{G}(\mathbf{V} \cup \mathbf{H})$, where $\mathbf{V}$ are observed and $\mathbf{H}$ are hidden variables, a latent projection $\mathcal{G}(\mathbf{V})$ is the following ADMG with a vertex set $\mathbf{V}$. An edge $A \to B$ exists in $\mathcal{G}(\mathbf{V})$ if there exists a directed path from $A$ to $B$ in $\mathcal{G}(\mathbf{V} \cup \mathbf{H})$ with all intermediate vertices in $\mathbf{H}$. Similarly, an edge $A \leftrightarrow B$ exists in $\mathcal{G}(\mathbf{V})$ if there exists a path without consecutive edges $\to \circ \leftarrow$ from $A$ to $B$ with the first edge on the path of the form $A \leftarrow$ and the last edge on the path of the form $\to B$, and all intermediate vertices on the path in $\mathbf{H}$. As an example of this operation, the graph in Fig. 1 (c) is the latent projection of Fig. 1 (a). Note that a variable pair in a latent projection $\mathcal{G}(\mathbf{V})$ may be connected by both a directed and a bidirected edge, and that multiple distinct hidden variable DAGs $\mathcal{G}_1(\mathbf{V} \cup \mathbf{H}_1)$ and $\mathcal{G}_2(\mathbf{V} \cup \mathbf{H}_2)$ may share the same latent projection ADMG.

## 4.2 The Nested Factorization

The nested factorization of $p(\mathbf{V})$ with respect to an ADMG $\mathcal{G}(\mathbf{V})$ is defined on *kernel* objects derived from $p(\mathbf{V})$ and *conditional ADMGs* derived from $\mathcal{G}(\mathbf{V})$. The derivations are via a fixing operation, which can be causally interpreted as a single application of the g-formula on a single variable (to either a graph or a kernel) to obtain another graph or another kernel.

### 4.2.1 Conditional Graphs And Kernels

A *kernel* $q_{\mathbf{V}}(\mathbf{V}|\mathbf{W})$ is a mapping from values in $\mathbf{W}$ to normalized densities over $\mathbf{V}$ [5]. In other words, kernels act like conditional distributions in the sense that $\sum_{\mathbf{v} \in \mathbf{V}} q_{\mathbf{V}}(\mathbf{v}|\mathbf{w}) = 1, \forall \mathbf{w} \in \mathbf{W}$. Conditioning and marginalization in kernels are defined in the usual way. For $\mathbf{A} \subseteq \mathbf{V}$, we define $q(\mathbf{A}|\mathbf{W}) \equiv \sum_{\mathbf{V} \setminus \mathbf{A}} q(\mathbf{V}|\mathbf{W})$ and $q(\mathbf{V} \setminus \mathbf{A}|\mathbf{A}, \mathbf{W}) \equiv q(\mathbf{V}|\mathbf{W})/q(\mathbf{A}|\mathbf{W})$.

A conditional acyclic directed mixed graph (CADMG) $\mathcal{G}(\mathbf{V}, \mathbf{W})$ is an ADMG in which the nodes are partitioned into $\mathbf{W}$, representing *fixed variables*, and $\mathbf{V}$, representing *random variables*. Variables in $\mathbf{W}$ have the property that only outgoing directed edges may be adjacent to them. Genealogic relationships generalize from ADMGs to CADMGs without change. Districts are defined to be subsets of $\mathbf{V}$ in a CADMG $\mathcal{G}$, e.g. no element of $\mathbf{W}$ is in any element of $\mathcal{D}(\mathcal{G})$.

### 4.2.2 Fixability and Fixing

A variable $V \in \mathbf{V}$ in a CADMG $\mathcal{G}$ is *fixable* if $\text{de}_{\mathcal{G}}(V) \cap \text{dis}_{\mathcal{G}}(V) = \emptyset$. In other words, $V$ is fixable if paths $V \leftrightarrow \cdots \leftrightarrow B$ and $V \to \cdots \to B$ do not *both* exist in $\mathcal{G}$ for any $B \in \mathbf{V} \setminus \{V\}$. Given a CADMG $\mathcal{G}(\mathbf{V}, \mathbf{W})$ and $V \in \mathbf{V}$ fixable in $\mathcal{G}$, the fixing operator $\phi_V(\mathcal{G})$ yields a new CADMG $\mathcal{G}'(\mathbf{V} \setminus \{V\}|\mathbf{W} \cup \{V\})$, where all edges with arrowheads into $V$ are removed, and all other edges in $\mathcal{G}$ are kept. Similarly, given a CADMG $\mathcal{G}(\mathbf{V}, \mathbf{W})$, a kernel $q_{\mathbf{V}}(\mathbf{V}|\mathbf{W})$, and $V \in \mathbf{V}$ fixable in $\mathcal{G}$, the fixing operator $\phi_V(q_{\mathbf{V}}; \mathcal{G})$ yields a new kernel $q'_{\mathbf{V} \setminus \{V\}}(\mathbf{V} \setminus \{V\}|\mathbf{W} \cup \{V\}) \equiv \frac{q_{\mathbf{V}}(\mathbf{V}|\mathbf{W})}{q_{\mathbf{V}}(V|\text{nd}_{\mathcal{G}}(V), \mathbf{W})}$.

Note that fixing is a probabilistic operation in which we divide a kernel by a conditional kernel. In some cases this operates as a conditioning operation, in other cases as a marginalization operation, and in yet other cases, as neither, depending on the structure of the kernel being divided.

For a set $\mathbf{S} \subseteq \mathbf{V}$ in a CADMG $\mathcal{G}$, if all vertices in $\mathbf{S}$ can be ordered into a sequence $\sigma_{\mathbf{S}} = \langle S_1, S_2, \dots \rangle$ such that $S_1$ is fixable in $\mathcal{G}$, $S_2$ in $\phi_{S_1}(\mathcal{G})$, etc., $\mathbf{S}$ is said to be *fixable* in $\mathcal{G}$, $\mathbf{V} \setminus \mathbf{S}$ is said to be *reachable* in $\mathcal{G}$, and $\sigma_{\mathbf{S}}$ is said to be *valid*. A reachable set $\mathbf{C}$ is said to be *intrinsic* if $\mathcal{G}_{\mathbf{C}}$ has a single district. We will define $\phi_{\sigma_{\mathbf{S}}}(\mathcal{G})$ and $\phi_{\sigma_{\mathbf{S}}}(q; \mathcal{G})$ via the usual function composition to yield operators that fix all elements in $\mathbf{S}$ in the order given by $\sigma_{\mathbf{S}}$.

The distribution $p(\mathbf{V})$ is said to obey the nested factorization for an ADMG $\mathcal{G}$ if there exists a set of kernels $\{q_{\mathbf{C}}(\mathbf{C} \mid \mathrm{pa}_{\mathcal{G}}(\mathbf{C})) \mid \mathbf{C} \text{ is intrinsic in } \mathcal{G}\}$ such that for every fixable $\mathbf{S}$, and any valid $\sigma_{\mathbf{S}}$, $\phi_{\sigma_{\mathbf{S}}}(p(\mathbf{V}); \mathcal{G}) = \prod_{\mathbf{D} \in \mathcal{D}(\phi_{\sigma_{\mathbf{S}}}(\mathcal{G}))} q_{\mathbf{D}}(\mathbf{D} \mid \mathrm{pa}_{\mathcal{G}}^s(\mathbf{D}))$. All valid fixing sequences for $\mathbf{S}$ yield the same CADMG $\mathcal{G}(\mathbf{V} \setminus \mathbf{S}, \mathbf{S})$, and if $p(\mathbf{V})$ obeys the nested factorization for $\mathcal{G}$, all valid fixing sequences for $\mathbf{S}$ yield the same kernel. As a result, for any valid sequence $\sigma$ for $\mathbf{S}$, we will redefine the operator $\phi_{\sigma}$, for both graphs and kernels, to be $\phi_{\mathbf{S}}$. In addition, it can be shown [16] that the above kernel set is characterized as:

$$\{q_{\mathbf{C}}(\mathbf{C} \mid \mathrm{pa}_{\mathcal{G}}(\mathbf{C})) \mid \mathbf{C} \text{ is intrinsic in } \mathcal{G}\} = \{\phi_{\mathbf{V} \setminus \mathbf{C}}(p(\mathbf{V}); \mathcal{G}) \mid \mathbf{C} \text{ is intrinsic in } \mathcal{G}\}.$$

Thus, we can re-express the above nested factorization as stating that for any fixable set $\mathbf{S}$, we have $\phi_{\mathbf{S}}(p(\mathbf{V}); \mathcal{G}) = \prod_{\mathbf{D} \in \mathcal{D}(\phi_{\mathbf{S}}(\mathcal{G}))} \phi_{\mathbf{V} \setminus \mathbf{D}}(p(\mathbf{V}); \mathcal{G})$. Since fixing is defined on CADMGs and kernels, the definition of nested Markov models generalizes in a straightforward way to a kernel $q(\mathbf{V}|\mathbf{W})$ being in the nested Markov model for a CADMG $\mathcal{G}(\mathbf{V}, \mathbf{W})$. This holds if for every $\mathbf{S}$ fixable in $\mathcal{G}(\mathbf{V}, \mathbf{W})$, $\phi_{\mathbf{S}}(q(\mathbf{V}|\mathbf{W}); \mathcal{G}) = \prod_{\mathbf{D} \in \mathcal{D}(\phi_{\mathbf{S}}(\mathcal{G}))} \phi_{\mathbf{V} \setminus \mathbf{D}}(q(\mathbf{V}|\mathbf{W}); \mathcal{G})$.

An important result in [16] states that if $p(\mathbf{V} \cup \mathbf{H})$ obeys the factorization for a DAG $\mathcal{G}$ with vertex set $\mathbf{V} \cup \mathbf{H}$, then $p(\mathbf{V})$ obeys the nested factorization for the latent projection ADMG $\mathcal{G}(\mathbf{V})$.

### 4.3 Identification in Hidden Variable Causal DAGs

For any disjoint subsets $\mathbf{Y}, \mathbf{A}$ of $\mathbf{V}$ in a latent projection $\mathcal{G}(\mathbf{V})$ representing a causal DAG $\mathcal{G}(\mathbf{V} \cup \mathbf{H})$, define $\mathbf{Y}^* \equiv \mathrm{an}_{\mathcal{G}(\mathbf{V})_{\mathbf{V} \setminus \mathbf{A}}}(\mathbf{Y})$. Then $p(\mathbf{Y}|\mathrm{do}(\mathbf{a}))$ is identified in $\mathcal{G}$ if *and only if* every set $\mathbf{D} \in \mathcal{D}(\mathcal{G}(\mathbf{V})_{\mathbf{Y}^*})$ is reachable (in fact, intrinsic). Moreover, if identification holds, we have [16]:

$$p(\mathbf{Y}|\mathrm{do}(\mathbf{a})) = \sum_{\mathbf{Y}^* \setminus \mathbf{Y}} \prod_{\mathbf{D} \in \mathcal{D}(\mathcal{G}(\mathbf{V})_{\mathbf{Y}^*})} \phi_{\mathbf{V} \setminus \mathbf{D}}(p(\mathbf{V}); \mathcal{G}(\mathbf{V}))|_{\mathbf{A}=\mathbf{a}}. \tag{2}$$

In other words, $p(\mathbf{Y}|\mathrm{do}(\mathbf{a}))$ is only identified if it can be expressed as a factorization, where every piece corresponds to a kernel associated with a set intrinsic in $\mathcal{G}(\mathbf{V})$. Moreover, no piece in this factorization contains elements of $\mathbf{A}$ as random variables, just as was the case in (1). In fact, (2) provides a concise formulation of the ID algorithm [27, 21] in terms of the nested Markov model in which the observed distribution in the causal problem lies. For a full proof, see [16].

## 5 Chain Graphs For Causal Inference With Dependent Data

We generalize causal models to represent settings with data dependence, specifically to cases where variables may exhibit stable but symmetric relationships. These may correspond to friendship ties in a social network, physical proximity, or rules of infectious disease spread. These stand in contrast to causal relationships which are also stable, but asymmetric. We represent settings with both of these kinds of relationships using causal CG models under the Lauritzen-Wermuth-Freydenburg (LWF) interpretation. Though there are alternative conceptions of chain graphs [2], we concentrate on LWF CGs here. This is because LWF CGs yield observed data distributions with smooth parameterizations. In addition, LWF CGs yield Markov properties where each unit's friends (and direct causes) screen the unit from other units in the network. This sort of independence is intuitively appealing in many network settings. Extensions of our results to other CG models are likely possible, but we leave them to future work.

LWF CGs were given a causal interpretation in [6]. In a causal CG, the distribution $p(\mathbf{B}| \mathrm{pa}_{\mathcal{G}}(\mathbf{B}))$ for each block $\mathbf{B}$ is determined via a computer program that implements a Gibbs sampler on variables $B \in \mathbf{B}$, where the conditional distribution $p(B|\mathbf{B} \setminus \{B\}, \mathrm{pa}_{\mathcal{G}}(\mathbf{B}))$ is determined via a structural equation of the form $f_B(\mathbf{B} \setminus \{B\}, \mathrm{pa}_{\mathcal{G}}(\mathbf{B}), \epsilon_B)$. This interpretation of $p(\mathbf{B}| \mathrm{pa}_{\mathcal{G}}(\mathbf{B}))$ allows the

implementation of a simple intervention operation do($b$). The operation sets $B$ to $b$ by replacing the line of the Gibbs sampler program that assigns $B$ to the value returned by $f_B(\mathbf{B} \setminus \{B\}, \mathrm{pa}_{\mathcal{G}}(\mathbf{B}), \epsilon_B)$ (given a new realization of $\epsilon_B$), with an assignment of $B$ to the value $b$. It was shown [6] that in a causal CG model, for any disjoint $\mathbf{Y}, \mathbf{A}$, $p(\mathbf{Y}|\mathrm{do}(\mathbf{a}))$ is identified by the CG version of the g-formula (1): $p(\mathbf{Y}|\mathrm{do}(\mathbf{a})) = \prod_{\mathbf{B} \in \mathcal{B}(\mathcal{G})} p(\mathbf{B} \setminus \mathbf{A} | \mathrm{pa}(\mathbf{B}), \mathbf{B} \cap \mathbf{A})|_{\mathbf{A} = \mathbf{a}}$.

In our example above, stable symmetric relationships inducing data dependence, represented by undirected edges, coexist with hidden variables. To represent causal inference in this setting, we generalize earlier developments for hidden variable causal DAG models to hidden variable causal CG models. Specifically, we first define a latent projection analogue called the segregated projection for a large class of hidden variable CGs using segregated graphs (SGs). We then define a factorization for SGs that generalizes the nested factorization and the CG factorization, and show that if a distribution $p(\mathbf{V} \cup \mathbf{H})$ factorizes given a CG $\mathcal{G}(\mathbf{V} \cup \mathbf{H})$ in the class, then $p(\mathbf{V})$ factorizes according to the segregated projection $\mathcal{G}(\mathbf{V})$. Finally, we derive identification theory for hidden variable CGs as a generalization of (2) that can be viewed as a truncated SG factorization.

## 5.1 Segregated Projections Of Latent Variable Chain Graphs

Fix a chain graph CG $\mathcal{G}$ and a vertex set $\mathbf{H}$ such that for all $H \in \mathbf{H}$, $H$ does not lie in $\mathbf{B} \cup \mathrm{pa}_{\mathcal{G}}(\mathbf{B})$, for any $\mathbf{B} \in \mathcal{B}^{nt}(\mathcal{G})$. We call such a set $\mathbf{H}$ *block-safe*.

**Definition 1** *Given a CG $\mathcal{G}(\mathbf{V} \cup \mathbf{H})$ and a block-safe set $\mathbf{H}$, define a segregated projection graph $\mathcal{G}(\mathbf{V})$ with a vertex set $\mathbf{V}$. Moreover, for any collider-free path from any two elements $V_1, V_2$ in $\mathbf{V}$, where all intermediate vertices are in $\mathbf{H}$, $\mathcal{G}(\mathbf{V})$ contains an edge with end points matching the path. That is, we have $V_1 \leftarrow \circ \ldots \circ \rightarrow V_2$ leads to the edge $V_1 \leftrightarrow V_2$, $V_1 \rightarrow \circ \ldots \circ \rightarrow V_2$ leads to the edge $V_1 \rightarrow V_2$, and in $\mathcal{G}(\mathbf{V})$.*

As an example, the SG in Fig. 1 (c) is a segregated projection of the hidden variable CG in Fig. 1 (a). While segregated graphs preserve conditional independence structure on the observed marginal of a CG for *any* $\mathbf{H}$ [20], we chose to further restrict the set $\mathbf{H}$ in order to ensure that the directed edges in the segregated projection retain an intuitive causal interpretation of edges in a latent projection [28]. That is, whenever $A \rightarrow B$ in a segregated projection, $A$ is a causal ancestor of $B$ in the underlying causal CG. SGs represent latent variable CGs, meaning that they allow causal systems that model feedback that leads to network structures, of the sort considered in [6], but simultaneously allow certain forms of unobserved confounding in such causal systems.

## 5.2 Segregated Factorization

The segregated factorization of an SG can be defined as a product of two kernels which themselves factorize, one in terms of a CADMG (a conditional graph with only directed and bidirected arrows), and another in terms of a *conditional chain graph (CCG)* $\mathcal{G}(\mathbf{V}, \mathbf{W})$, a CG with the property that the only type of edge adjacent to any element $W$ of $\mathbf{W}$ is a directed edge out of $W$. A kernel $q(\mathbf{V}|\mathbf{W})$ is said to be Markov relative to the CCG $\mathcal{G}(\mathbf{V}, \mathbf{W})$ if $q(\mathbf{V}|\mathbf{W}) = Z(\mathbf{W})^{-1} \prod_{\mathbf{B} \in \mathcal{B}(\mathcal{G})} q(\mathbf{B}| \mathrm{pa}_{\mathcal{G}}(\mathbf{B}))$, and $q(\mathbf{B}| \mathrm{pa}_{\mathcal{G}}(\mathbf{B})) = Z(\mathrm{pa}_{\mathcal{G}}(\mathbf{B}))^{-1} \prod_{\mathbf{C} \in \mathcal{C}((\mathcal{G}_{\mathbf{B} \cup \mathrm{pa}_{\mathcal{G}}(\mathbf{B})})^a); \mathbf{C} \not\subseteq \mathrm{pa}_{\mathcal{G}}(\mathbf{B})} \psi_{\mathbf{C}}(\mathbf{C})$, for each $\mathbf{B} \in \mathcal{B}(\mathcal{G})$.

We now show, given $p(\mathbf{V})$ and an SG $\mathcal{G}(\mathbf{V})$, how to construct the appropriate CADMG and CCG, and the two corresponding kernels. Given a SG $\mathcal{G}$, let *district variables* $\mathbf{D}^*$ be defined as $\bigcup_{\mathbf{D} \in \mathcal{D}(\mathcal{G})} \mathbf{D}$, and let *block variables* $\mathbf{B}^*$ be defined as $\bigcup_{\mathbf{B} \in \mathcal{B}^{nt}(\mathcal{G})} \mathbf{B}$. Since $\mathcal{D}(\mathcal{G})$ and $\mathcal{B}^{nt}(\mathcal{G})$ partition $\mathbf{V}$ in a SG, $\mathbf{B}^*$ and $\mathbf{D}^*$ partition $\mathbf{V}$ as well. Let the induced CADMG $\mathcal{G}^d$ of a SG $\mathcal{G}$ be the graph containing the vertex sets $\mathbf{D}^*$ as $\mathbf{V}$ and $\mathrm{pa}_{\mathcal{G}}^s(\mathbf{D}^*)$ as $\mathbf{W}$, and which inherits all edges in $\mathcal{G}$ between $\mathbf{D}^*$, and all directed edges from $\mathrm{pa}_{\mathcal{G}}^s(\mathbf{D}^*)$ to $\mathbf{D}^*$ in $\mathcal{G}$. Similarly, let the induced CCG $\mathcal{G}^b$ of $\mathcal{G}$ be the graph containing the vertex set $\mathbf{B}^*$ as $\mathbf{V}$ and $\mathrm{pa}_{\mathcal{G}}^s(\mathbf{B}^*)$ as $\mathbf{W}$, and which inherits all edges in $\mathcal{G}$ between $\mathbf{B}^*$, and all directed edges from $\mathrm{pa}_{\mathcal{G}}(\mathbf{B}^*)$ to $\mathbf{B}^*$. We say that $p(\mathbf{V})$ obeys the factorization of a SG $\mathcal{G}(\mathbf{V})$ if $p(\mathbf{V}) = q(\mathbf{D}^*| \mathrm{pa}_{\mathcal{G}}^s(\mathbf{D}^*))q(\mathbf{B}^*| \mathrm{pa}_{\mathcal{G}}^s(\mathbf{B}^*))$, $q(\mathbf{B}^*| \mathrm{pa}_{\mathcal{G}}^s(\mathbf{B}^*))$ is Markov relative to the CCG $\mathcal{G}^b$, and $q(\mathbf{D}^*| \mathrm{pa}_{\mathcal{G}}^s(\mathbf{D}^*))$ is in the nested Markov model of the CADMG $\mathcal{G}^d$.

The following theorem gives the relationship between a joint distribution that factorizes given a hidden variable CG $\mathcal{G}$, its marginal distribution, and the corresponding segregated factorization. This

theorem is a generalization of the result proven in [16] relating hidden variable DAGs and latent projection ADMGs. The proof is deferred to the appendix.

**Theorem 1** *If $p(\mathbf{V} \cup \mathbf{H})$ obeys the CG factorization relative to $\mathcal{G}(\mathbf{V} \cup \mathbf{H})$, and $\mathbf{H}$ is block-safe then $p(\mathbf{V})$ obeys the segregated factorization relative to the segregated projection $\mathcal{G}(\mathbf{V})$.*

## 6 A Complete Identification Algorithm for Latent Variable Chain Graphs

With Theorem 1 in hand, we are ready to characterize general non-parametric identification of interventional distributions in hidden variable causal chain graph models, where hidden variables form a block-safe set. This result can be viewed on the one hand as a generalization of the CG g-formula derived in [6], and on the other hand as a generalization of the ID algorithm (2).

**Theorem 2** *Assume $\mathcal{G}(\mathbf{V} \cup \mathbf{H})$ is a causal CG, where $\mathbf{H}$ is block-safe. Fix disjoint subsets $\mathbf{Y}, \mathbf{A}$ of $\mathbf{V}$. Let $\mathbf{Y}^* = \mathrm{ant}_{\mathcal{G}(\mathbf{V})_{\mathbf{V} \setminus \mathbf{A}}} \mathbf{Y}$. Then $p(\mathbf{Y}|do(\mathbf{a}))$ is identified from $p(\mathbf{V})$ if and only if every element in $\mathcal{D}(\widetilde{\mathcal{G}}^d)$ is reachable in $\mathcal{G}^d$, where $\widetilde{\mathcal{G}}^d$ is the induced CADMG of $\mathcal{G}(\mathbf{V})_{\mathbf{Y}^*}$.*

*Moreover, if $p(\mathbf{Y}|do(\mathbf{a}))$ is identified, it is equal to*

$$\sum_{\mathbf{Y}^* \setminus \mathbf{Y}} \left[ \prod_{\mathbf{D} \in \mathcal{D}(\widetilde{\mathcal{G}}^d)} \phi_{\mathbf{D}^* \setminus \mathbf{D}}(q(\mathbf{D}^* | \mathrm{pa}_{\mathcal{G}(\mathbf{V})}(\mathbf{D}^*)); \mathcal{G}^d) \right] \left[ \prod_{\mathbf{B} \in \mathcal{B}(\widetilde{\mathcal{G}}^b)} p(\mathbf{B} \setminus \mathbf{A} | \mathrm{pa}_{\mathcal{G}(\mathbf{V})_{\mathbf{Y}^*}}(\mathbf{B}), \mathbf{B} \cap \mathbf{A}) \right] \Bigg|_{\mathbf{A}=\mathbf{a}}$$

*where $q(\mathbf{D}^* | \mathrm{pa}_{\mathcal{G}(\mathbf{V})}(\mathbf{D}^*)) = p(\mathbf{V})/(\prod_{\mathbf{B} \in \mathcal{B}^{nt}(\mathcal{G}(\mathbf{V}))} p(\mathbf{B} | \mathrm{pa}_{\mathcal{G}(\mathbf{V})}(\mathbf{B})))$, and $\widetilde{\mathcal{G}}^b$ is the induced CCG of $\mathcal{G}(\mathbf{V})_{\mathbf{Y}^*}$.*

To illustrate the application of this theorem, consider the SG $\mathcal{G}$ in Fig. 1 (c), where we are interested in $p(Y_2|do(a_1, a_2))$. It is easy to see that $\mathbf{Y}^* = \{C_1, C_2, M_1, M_2, Y_2\}$ (see $\mathcal{G}_{\mathbf{Y}^*}$ in Fig. 1 (d)) with $\mathcal{B}(\mathcal{G}_{\mathbf{Y}^*}) = \{\{M_1, M_2\}\}$ and $\mathcal{D}(\mathcal{G}_{\mathbf{Y}^*}) = \{\{C_1\}, \{C_2\}, \{Y_2\}\}$. The chain graph factor of the factorization in Theorem 2 is $p(M_1, M_2|A_1 = a_1, A_2, C_1, C_2)$. Note that this expression further factorizes according to the (second level) undirected factorization of blocks in a CCG. For the three district factors $\{C_1\}, \{C_2\}, \{Y_2\}$ in Fig. 1 (d), we must fix variables in three different sets $\{C_2, A_1, A_2, Y_1, Y_2\}, \{C_1, A_1, A_2, Y_1, Y_2\}, \{C_1, C_2, A_1, Y_1, A_2\}$ in $\mathcal{G}^d$, shown in Fig. 1 (e). We defer the full derivation involving the fixing operator to the supplementary material. The resulting identifying functional for $p(Y_2|do(a_1, a_2))$ is:

$$\sum_{\{C_1, C_2, M_1, M_2\}} p(M_1, M_2|a_1, a_2, C_1, C_2) \sum_{A_2} p(Y_2|a_1, A_2, M_2, C_2)p(A_2|C_2)p(C_1)p(C_2) \tag{3}$$

## 7 Experiments

We now illustrate how identified functionals given by Theorem 2 may be estimated from data. Specifically we consider network average effects (N.E.), the network analogue of the average causal effect (ACE), as defined in [3]:

$$\mathrm{NE}^i(\mathbf{a}_{-i}) = \frac{1}{N} \sum_i E[Y_i(A_i = 1, \mathbf{A}_{-1} = 1)] - E[Y_i(A_i = 0, \mathbf{A}_{-i} = 0)]$$

in our article sharing example described in section 2, and shown in simplified form (for two units) in Fig. 1 (a). The experiments and results we present here generalize easily to other network effects such as direct and spillover effects [3], although we do not consider this here in the interests of space. For purposes of illustration we consider a simple setting where the social network is a 3-regular graph, with networks of size $N = [400, 800, 1000, 2000]$. Under the hidden variable CG model we described in section 2, the above effect is identified by a functional which generalizes (3) from a network of size 2 to a larger network. Importantly, since we assume a single connected network of $M$ variables, we are in the *full interference setting* where only a single sample from $p(M_1, \ldots M_N|A_1, \ldots, A_N, C_1, \ldots, C_N)$ is available. This means that while the standard maximum likelihood plug-in estimation strategy is possible for models for $Y_i$ and $A_i$ in (3), the strategy does not work for the model for $M$. Instead, we adapt the auto-g-computation approach based on the pseudo-likelihood and coding estimators proposed in [24], which is appropriate for full interference

settings with a Markov property given by a CG, as part of our estimation procedure. Note that the approach in [24] was applied for a special case of the set of causal models considered here, in particular those with no unmeasured confounding. Here we use the same approach for estimating general functionals in models that may include unobserved confounders between treatments and outcomes. In fact, our example model is analogous to the model in [24], in the same way that the front-door criterion is to the backdoor criterion in causal inference under the assumption of iid data [12].

Our detailed estimation strategy, along with a more detailed description of our results, is described in the appendix. We performed $1000$ bootstrap samples of the $4$ different networks. Since calculating the true causal effects is intractable even if true model parameters are known, we calculate the approximate 'ground truth' for each intervention by sampling from our data generating process under the intervention $5$ times and averaging the relevant effect. We calculated the (approximation of) the bias of each effect by subtracting the estimate from the 'ground truth.' The 'ground truth' network average effects range from $-.453$ to $-.456$. As shown in Tables 1 and 2, both estimators recover the ground truth effect with relatively small bias. Estimators for effects which used the pseudo-likelihood estimator for $M$ generally have lower variance than those that used the coding estimator for $M$, which is expected due to the greater efficiency of the former. This behavior was also observed in [24]. In both estimators, bias decreases with network size. This is also expected intuitively, although detailed asymptotic theory for statistical inference in networks is currently an open problem, due to dependence of samples.

| 95% Confidence Intervals of Bias of Network Average Effects | | | | | |
|---|---|---|---|---|---|
| | $N$ | 400 | 800 | 1000 | 2000 |
| Estimator | Coding | (-.157, .103) | (-.129, .106) | (-.100, .065) | (-.086, .051) |
| | Pseudo | (-.133, .080) | (-.099, .089) | (-.116, .074) | (-.070, .041) |

Table 1: 95% confidence intervals for the bias of each estimating method for the network average effects. All intervals cover the approximated ground truth since they include $0$

| Bias of Network Average Effects | | | | | |
|---|---|---|---|---|---|
| | $N$ | 400 | 800 | 1000 | 2000 |
| Estimator | Coding | -.000 (.060) | -.020 (.051) | -.024 (.052) | -.022 (.034) |
| | Pseudo | .006 (.052) | -.023 (.042) | -.023 (.042) | -.021 (.026) |

Table 2: The biases of each estimating method for the network average effects. Standard deviation of the bias of each estimate is given in parentheses.

# 8   Conclusion

In this paper, we generalized existing non-parametric identification theory for hidden variable causal DAG models to hidden variable causal chain graph models, which can represent both causal relationships, and stable symmetric relationships that induce data dependence. Specifically, we gave a representation of all identified interventional distributions in such models as a truncated factorization associated with *segregated graphs*, mixed graphs containing directed, undirected, and bidirected edges which represent marginals of chain graphs.

We also demonstrated how statistical inference may be performed on identifiable causal parameters, by adapting a combination of maximum likelihood plug in estimation, and methods based on coding and pseudo-likelihood estimators that were adapted for full interference problems in [24]. We illustrated our approach with an example of calculating the effect of community membership on article sharing if the effect of the former on the latter is mediated by a complex social network of units inducing full dependence.

# 9   Acknowledgements

The second author would like to thank the American Institute of Mathematics for supporting this research via the SQuaRE program. This project is sponsored in part by the National Institutes of

Health grant R01 AI127271-01 A1, the Office of Naval Research grant N00014-18-1-2760 and the Defense Advanced Research Projects Agency (DARPA) under contract HR0011-18-C-0049. The content of the information does not necessarily reflect the position or the policy of the Government, and no official endorsement should be inferred.

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
