[Supplementary Material · Supplementary Material for Identification and Estimation of Causal Effects from Dependent Data.pdf]

# Supplementary Material to Identification and Estimation Of Causal Effects from Dependent Data

**Eli Sherman**
Department of Computer Science
Johns Hopkins University
Baltimore, MD 21218
esherman@jhu.edu

**Ilya Shpitser**
Department of Computer Science
Johns Hopkins University
Baltimore, MD 21218
ilyas@cs.jhu.edu

## 1  Proofs

**Theorem 1** *If $p(\mathbf{V} \cup \mathbf{H})$ obeys the CG factorization relative to $\mathcal{G}(\mathbf{V} \cup \mathbf{H})$, and $\mathbf{H}$ is block-safe then $p(\mathbf{V})$ obeys the segregated factorization relative to the segregated projection $\mathcal{G}(\mathbf{V})$.*

*Proof:* Assume the premise of the theorem. Then, $p(\mathbf{O} \cup \mathbf{H}) = \prod_{\mathbf{B} \in \mathcal{B}(\mathcal{G})} p(\mathbf{B}|\operatorname{pa}_{\mathcal{G}}(\mathbf{B}))$.

For every $\mathbf{D} \in \mathcal{D}(\mathcal{G}(\mathbf{V}))$, let $\mathbf{H_D} \equiv \mathbf{H} \cap \operatorname{an}_{\mathcal{G}_{\mathbf{D} \cup \mathbf{H}}}(\mathbf{D})$. Then $p(\mathbf{V})$ is equal to

$$
\sum_{\mathbf{H}} \left( \prod_{\mathbf{B} \in \mathcal{B}^{nt}(\mathcal{G})} p(\mathbf{B}|\operatorname{pa}_{\mathcal{G}}(\mathbf{B})) \right) \left( \prod_{\{B\} \notin \mathcal{B}^{nt}(\mathcal{G})} p(B|\operatorname{pa}_{\mathcal{G}}(B)) \right)
$$

$$
= \left( \prod_{\mathbf{B} \in \mathcal{B}^{nt}(\mathcal{G})} p(\mathbf{B}|\operatorname{pa}_{\mathcal{G}}(\mathbf{B})) \right) \prod_{\mathbf{D} \in \mathcal{D}(\mathcal{G}(\mathbf{V}))} \sum_{\mathbf{H_D}} \left( \prod_{B \in \mathbf{D}} p(B|\operatorname{pa}_{\mathcal{G}}(B)) \right)
$$

$$
= \left( \prod_{\mathbf{B} \in \mathcal{B}^{nt}(\mathcal{G})} p(\mathbf{B}|\operatorname{pa}_{\mathcal{G}}(\mathbf{B})) \right) \prod_{\mathbf{D} \in \mathcal{D}(\mathcal{G}(\mathbf{V}))} q(\mathbf{D}|\operatorname{pa}^{s}_{\mathcal{G}(\mathbf{V})}(\mathbf{D}))
$$

$$
= q(\mathbf{B}^{*}|\operatorname{pa}_{\mathcal{G}(\mathbf{V})}(\mathbf{B}^{*})) q(\mathbf{D}^{*}|\operatorname{pa}^{s}_{\mathcal{G}(\mathbf{V})}(\mathbf{D}^{*})).
$$

The fact that $q(\mathbf{B}^{*}|\operatorname{pa}_{\mathcal{G}(\mathbf{V})}(\mathbf{B}^{*}))$ factorizes according to the CCG $\mathcal{G}^{b}$ follows by construction.

Let $\widetilde{\mathbf{B}} \equiv \{B \in \mathbf{V} \cup \mathbf{H} \mid \{B\} \notin \mathcal{B}^{nt}(\mathcal{G})\}$. Then

$$
q(\widetilde{\mathbf{B}}|\operatorname{pa}^{s}_{\mathcal{G}}(\widetilde{\mathbf{B}})) = \prod_{B:\{B\} \notin \mathcal{B}^{nt}(\mathcal{G})} p(B|\operatorname{pa}_{\mathcal{G}}(B))
$$

factorizes according to the CADMG (in fact a conditional DAG) $\mathcal{G}(\widetilde{\mathbf{B}}, \operatorname{pa}^{s}_{\mathcal{G}}(\widetilde{\mathbf{B}}))$ obtained from $\mathcal{G}(\mathbf{V} \cup \mathbf{H})$ by making all elements in $\operatorname{pa}^{s}_{\mathcal{G}}(\widetilde{\mathbf{B}})$ fixed, and all elements $\widetilde{\mathbf{B}}$ random, keeping all edges among $\widetilde{\mathbf{B}}$ in $\mathcal{G}$, and all outgoing directed edges from $\operatorname{pa}^{s}_{\mathcal{G}}(\widetilde{\mathbf{B}})$ to $\widetilde{\mathbf{B}}$ in $\mathcal{G}$. The fact that $q(\mathbf{D}^{*}|\operatorname{pa}_{\mathcal{G}(\mathbf{V})}(\mathbf{D}^{*}))$ factorizes according $\mathcal{G}^{d}$, the latent projection CADMG obtained from $\mathcal{G}(\widetilde{\mathbf{B}}, \operatorname{pa}^{s}_{\mathcal{G}}(\widetilde{\mathbf{B}}))$ by treating $\mathbf{H}$ as hidden variables now follows by the inductive application of Lemmas 46 and 49 in [2] to $q(\widetilde{\mathbf{B}}|\operatorname{pa}^{s}_{\mathcal{G}}(\widetilde{\mathbf{B}}))$ and $\mathcal{G}(\widetilde{\mathbf{B}}, \operatorname{pa}^{s}_{\mathcal{G}}(\widetilde{\mathbf{B}}))$. □

**Theorem 2** *Assume $\mathcal{G}(\mathbf{V} \cup \mathbf{H})$ is a causal CG, where $\mathbf{H}$ is block-safe. Fix disjoint subsets $\mathbf{Y}$, $\mathbf{A}$ of $\mathbf{V}$. Let $\mathbf{Y}^{*} = \operatorname{ant}_{\mathcal{G}(\mathbf{V})_{\mathbf{V} \setminus \mathbf{A}}} \mathbf{Y}$. Then $p(\mathbf{Y}|do(\mathbf{a}))$ is identified from $p(\mathbf{V})$ if and only if every element in $\mathcal{D}(\widetilde{\mathcal{G}}^{d})$ is reachable in $\mathcal{G}^{d}$, where $\widetilde{\mathcal{G}}^{d}$ is the induced CADMG of $\mathcal{G}(\mathbf{V})_{\mathbf{Y}^{*}}$.*

*Moreover, if $p(\mathbf{Y}|do(\mathbf{a}))$ is identified, it is equal to*

$$\sum_{\mathbf{Y}^*\setminus\mathbf{Y}} \left[\prod_{\mathbf{D}\in\mathcal{D}(\widetilde{\mathcal{G}}^d)} \phi_{\mathbf{D}^*\setminus\mathbf{D}}(q(\mathbf{D}^*|\operatorname{pa}_{\mathcal{G}(\mathbf{V})}(\mathbf{D}^*)); \mathcal{G}^d)\right]\left[\prod_{\mathbf{B}\in\mathcal{B}(\widetilde{\mathcal{G}}^b)} p(\mathbf{B}\setminus\mathbf{A}|\operatorname{pa}_{\mathcal{G}(\mathbf{V})_{\mathbf{Y}^*}}(\mathbf{B}), \mathbf{B}\cap\mathbf{A})\right]\Bigg|_{\mathbf{A}=\mathbf{a}}$$
(1)

*where $q(\mathbf{D}^*|\operatorname{pa}_{\mathcal{G}(\mathbf{V})}(\mathbf{D}^*)) = p(\mathbf{V})/(\prod_{\mathbf{B}\in\mathcal{B}^{nt}(\mathcal{G}(\mathbf{V}))} p(\mathbf{B}|\operatorname{pa}_{\mathcal{G}(\mathbf{V})}(\mathbf{B})))$, and $\widetilde{\mathcal{G}}^d$ is the induced CCG of $\mathcal{G}(\mathbf{V})_{\mathbf{Y}^*}$.*

*Proof:* We proceed by proving a series of subclaims.

**Claim 1**: *If $p(\mathbf{O})$ obeys the segregated factorization relative to $\mathcal{G}(\mathbf{O})$, then $p(\mathbf{A})$ obeys the segregated factorization relative to $\mathcal{G}(\mathbf{O})_{\mathbf{A}}$ for any subset $\mathbf{A}\subseteq\mathbf{O}$ anterial in $\mathcal{G}(\mathbf{O})$. A set $\mathbf{A}$ is anterial if, whenever $X\in\mathbf{A}$, $\operatorname{ant}_{\mathcal{G}}(X)\subseteq\mathbf{A}$.*

We show this by induction. Assume $p(\mathbf{O})$ obeys the segregated factorization relative to $\mathcal{G}(\mathbf{O})$, and $\mathbf{A}$ consists of all elements in $\mathbf{O}$ other than those in $\mathbf{B}\in\mathcal{B}^{nt}(\mathcal{G}(\mathbf{O}))$. Then by writing $p(\mathbf{A})=\sum_{\mathbf{B}} p(\mathbf{O})$ as a segregated factorization for $p(\mathbf{O})$, we note that the nested factorization remains unchanged by the marginalization, and the block factorization remains unchanged, except the factor corresponding to $\mathbf{B}$ is removed.

Similarly, assume $p(\mathbf{O})$ obeys the segregated factorization relative to $\mathcal{G}(\mathbf{O})$, and $\mathbf{A}$ consists of all elements in $\mathbf{O}$ other than some element $B$ not in any $\mathbf{B}\in\mathcal{B}^{nt}(\mathcal{G}(\mathbf{O}))$ such that $\operatorname{ch}_{\mathcal{G}}(B)$ is empty. Then by writing $p(\mathbf{A})=\sum_{\mathbf{B}} p(\mathbf{O})$ as a segregated factorization for $p(\mathbf{O})$, we note that the block factorization remains unchanged by the marginalization, and the kernel

$$q(\mathbf{B}^*\setminus\{B\}\mid\operatorname{pa}^s_{\mathcal{G}(\mathbf{O})}(\mathbf{B}^*)) = \sum_B \frac{p(\mathbf{V})}{\prod_{\mathbf{B}\in\mathcal{B}^{nt}(\mathcal{G}(\mathbf{V}))} p(\mathbf{B}|\operatorname{pa}_{\mathcal{G}(\mathbf{V})}(\mathbf{B}))}$$

is nested Markov relative to the CADMG $\tilde{\mathcal{G}}(\mathbf{O})^d$ obtained from $\mathcal{G}(\mathbf{O})^d$ by removing $B$ and all edges adjacent to $B$. To see this, note that reachable sets in $\tilde{\mathcal{G}}(\mathbf{O})^d$ are a strict subset of reachable sets in $\mathcal{G}(\mathbf{O})^d$, since $B$ is fixable in $\mathcal{G}(\mathbf{O})^d$, and moreover all kernels corresponding to reachable sets in $\tilde{\mathcal{G}}(\mathbf{O})^d$ may be obtained from $q(\mathbf{B}^*\mid\operatorname{pa}^s_{\mathcal{G}(\mathbf{O})}(\mathbf{B}^*))$ by marginalizing $B$ first, and applying the fixing operator to remaining variables in $\mathcal{B}^*\setminus\{B\}$. As a result, the nested global Markov property for the former graph is implied by the nested global Markov property of the latter graph, proving our claim.

**Claim 2**: *The algorithm specified by the equation (1) is sound for identification of $p(\mathbf{Y}|do(\mathbf{a}))$.*

Per claim 1, without loss of generality assume $\mathbf{Y}$ has no children in $\mathcal{G}(\mathbf{O})$. Consider the chain graph g-formula:

$$p(\mathbf{Y}(\mathbf{a})) = \prod_{\mathbf{B}\in\mathcal{B}(\mathcal{G}(\mathbf{O}\cup\mathbf{H}))} p(\mathbf{B}\setminus\mathbf{A}|\operatorname{pa}_{\mathcal{G}}(\mathbf{B}), \mathbf{B}\cap\mathbf{A})|_{\mathbf{A}=\mathbf{a}}.$$

We can decompose this into factors relating to the non-trivial blocks and districts in the graph:

$$p(\mathbf{Y}(\mathbf{a})) = \prod_{\mathbf{B}\in\mathcal{B}^{nt}(\mathcal{G}(\mathbf{O}\cup\mathbf{H}))} p(\mathbf{B}\setminus\mathbf{A}|\operatorname{pa}_{\mathcal{G}}(\mathbf{B}), \mathbf{B}\cap\mathbf{A})|_{\mathbf{A}=\mathbf{a}}$$
$$\times \prod_{\mathbf{D}\in\mathcal{D}(\mathcal{G}(\mathbf{O}\cup\mathbf{H}))} p(\mathbf{D}\setminus\mathbf{A}|\operatorname{pa}_{\mathcal{G}}(\mathbf{D}), \mathbf{D}\cap\mathbf{A})|_{\mathbf{A}=\mathbf{a}}.$$

Since $\mathbf{H}$ is block-safe, the factors in the first term – those that correspond to non-trivial blocks – are the same in the segregated graph as in the original chain graph and thus we can re-write the above as:

$$p(\mathbf{Y}(\mathbf{a})) = \prod_{\mathbf{B}\in\mathcal{B}^{nt}(\mathcal{G}_{\mathbf{Y}^*})} p(\mathbf{B}\setminus\mathbf{A}|\operatorname{pa}_{\mathcal{G}}(\mathbf{B}), \mathbf{B}\cap\mathbf{A})|_{\mathbf{A}=\mathbf{a}}$$
$$\times \prod_{\mathbf{D}\in\mathcal{D}(\mathcal{G}(\mathbf{O}\cup\mathbf{H}))} p(\mathbf{D}\setminus\mathbf{A}|\operatorname{pa}_{\mathcal{G}}(\mathbf{D}), \mathbf{D}\cap\mathbf{A})|_{\mathbf{A}=\mathbf{a}}.$$

Meanwhile the factors in the second term describe a kernel $q(\mathbf{D}^*|\operatorname{pa}_{\mathcal{G}_{(\mathbf{O}\cup\mathbf{H})}}(\mathbf{D}^*))$ associated with a CADG $\mathcal{G}(\mathbf{O}\cup\mathbf{H}, \mathbf{B}^*)$ which we can manipulate to obtain the desired result by following the argument in the proof of Theorem 60 in [2].

Let $\mathbf{A}^* = \mathbf{O} \setminus \mathbf{Y}^* \supseteq \mathbf{A}$. By the global Markov property of conditional DAGs (CDAGs) proven in [2], $p(\mathbf{Y}^*|do_{\mathcal{G}(\mathbf{O}\cup\mathbf{H},\mathbf{B}^*)}(\mathbf{a})) = p(\mathbf{Y}^*|do_{\mathcal{G}(\mathbf{O}\cup\mathbf{H},\mathbf{B}^*)}(\mathbf{a}^*))$.

Let $\mathcal{G}^*((\mathbf{O}\setminus\mathbf{A}^*)\cup\mathbf{H},\mathbf{B}^*\cup\mathbf{A}^*) = \phi_{\mathbf{A}^*}(\mathcal{G}(\mathbf{O}\cup\mathbf{H},\mathbf{B}^*))$. Let $\sigma_\mathbf{H}$ denote the *latent projection operation* such that $\sigma_\mathbf{H}(\mathcal{G}(\mathbf{O}\cup\mathbf{H})) = \mathcal{G}(\mathbf{O})$. Then, by commutativity of $\sigma_\mathbf{H}$ and the fixing operator (Corollary 53 in [2]), $\sigma_\mathbf{H}(\phi_{\mathbf{A}^*}(\mathcal{G}(\mathbf{O}\cup\mathbf{H},\mathbf{B}^*))) = \phi_{\mathbf{A}^*}(\sigma_\mathbf{H}(\mathcal{G}(\mathbf{O}\cup\mathbf{H},\mathbf{B}^*))) = \mathcal{G}^*(\mathbf{Y}^*,\mathbf{B}^*\cup\mathbf{A}^*)$. By definition of induced subgraphs, $\mathcal{G}(\mathbf{O},\mathbf{B}^*)_{\mathbf{Y}^*} = (\phi_{\mathbf{A}^*}(\mathcal{G}(\mathbf{O},\mathbf{B}^*)))_{\mathbf{Y}^*}$. By these two equalities, we have $\mathcal{G}(\mathbf{O},\mathbf{B}^*)_{\mathbf{Y}^*} = \mathcal{G}^*(\mathbf{O},\mathbf{B}^*\cup\mathbf{A}^*)_{\mathbf{Y}^*}$ and thus $\mathcal{D}(\mathcal{G}(\mathbf{O},\mathbf{B}^*)_{\mathbf{Y}^*}) = \mathcal{D}(\mathcal{G}^*(\mathbf{Y}^*,\mathbf{B}^*\cup\mathbf{A}^*))$.

For each $\mathbf{D}\in\mathcal{D}(\mathcal{G}^*(\mathbf{Y}^*,\mathbf{B}^*\cup\mathbf{A}^*))$, let $\mathbf{H_D}\equiv\mathbf{H}\cap\mathrm{an}_{\mathcal{G}(\mathbf{O}\cup\mathbf{H},\mathbf{B}^*)_{\mathbf{D}\cup\mathbf{H}}}(\mathbf{D})$ and $\mathbf{H}^*\equiv\bigcup_{\mathbf{D}\in\mathcal{D}(\mathcal{G}^*(\mathbf{Y}^*,\mathbf{B}^*\cup\mathbf{A}^*))}\mathbf{H_D}$. Then, by construction, if $\mathbf{D},\mathbf{D}'\in\mathcal{D}(\mathcal{G}^*(\mathbf{Y}^*,\mathbf{B}^*\cup\mathbf{A}^*))$ and $\mathbf{D}\neq\mathbf{D}'$ then $\mathbf{H_D}\cap\mathbf{H_{D'}}=\emptyset$. Additionally, for all $\mathbf{D}\in\mathcal{D}(\mathcal{G}^*(\mathbf{Y}^*,\mathbf{B}^*\cup\mathbf{A}^*))$, it is the case that $\mathrm{pa}_{\mathcal{G}(\mathbf{O}\cup\mathbf{H},\mathbf{B}^*)}(\mathbf{D}\cup\mathbf{H_D})\cap\mathbf{H}^*=\mathbf{H_D}$. And $\mathbf{Y}^*\cup\mathbf{H}^*$ is ancestral in $\mathcal{G}(\mathbf{O}\cup\mathbf{H},\mathbf{B}^*)$ which implies that if $v\in\mathbf{Y}^*\cup\mathbf{H}^*$, then $\mathrm{pa}_{\mathcal{G}(\mathbf{O}\cup\mathbf{H},\mathbf{B}^*)}(v)\cap\mathbf{H}\subseteq\mathbf{H}^*$.

By the DAG g-formula and the above features of the construction,

$$
\begin{aligned}
p(\mathbf{Y}^*&|do_{\mathcal{G}(\mathbf{O}\cup\mathbf{H},\mathbf{B}^*)}(\mathbf{a}^*))\\
&=\sum_\mathbf{H}\prod_{v\in(\mathbf{H}\cup\mathbf{Y}^*)}p(v|\,\mathrm{pa}_{\mathcal{G}(\mathbf{O}\cup\mathbf{H},\mathbf{B}^*)}(v))\\
&=\sum_{\mathbf{H}^*}\prod_{v\in(\mathbf{H}^*\cup\mathbf{Y}^*)}p(v|\,\mathrm{pa}_{\mathcal{G}(\mathbf{O}\cup\mathbf{H},\mathbf{B}^*)}(v))\cdot\sum_{\mathbf{H}\setminus\mathbf{H}^*}\prod_{v\in(\mathbf{H}\setminus\mathbf{H}^*)}p(v|\,\mathrm{pa}_{\mathcal{G}(\mathbf{O}\cup\mathbf{H},\mathbf{B}^*)}(v))\\
&=\sum_{\mathbf{H}^*}\prod_{\mathbf{D}\in\mathcal{D}(\mathcal{G}^*(\mathbf{Y}^*,\mathbf{A}^*\cup\mathbf{B}^*))}\prod_{v\in(\mathbf{D}\cup\mathbf{H_D})}p(v|\,\mathrm{pa}_{\mathcal{G}(\mathbf{O}\cup\mathbf{H},\mathbf{B}^*)}(v))\\
&=\prod_{\mathbf{D}\in\mathcal{D}(\mathcal{G}^*(\mathbf{Y}^*,\mathbf{A}^*\cup\mathbf{B}^*))}\left(\sum_{\mathbf{H_D}}\prod_{v\in(\mathbf{D}\cup\mathbf{H_D})}p(v|\,\mathrm{pa}_{\mathcal{G}(\mathbf{O}\cup\mathbf{H},\mathbf{B}^*)}(v))\right).
\end{aligned}\tag{2}
$$

For any district $\mathbf{D}\in\mathcal{D}(\mathcal{G}^*(\mathbf{Y}^*,\mathbf{B}^*\cup\mathbf{A}^*))$,

$$
\begin{aligned}
\sum_{\mathbf{H_D}}&\prod_{v\in\mathbf{D}\cup\mathbf{H_D}}p(v|\,\mathrm{pa}_{\mathcal{G}(\mathbf{O}\cup\mathbf{H},\mathbf{B}^*)}(v))\\
&=\sum_{\mathbf{H_D}}\prod_{v\in(\mathbf{D}\cup\mathbf{H_D})}p(v|\,\mathrm{pa}_{\mathcal{G}(\mathbf{O}\cup\mathbf{H},\mathbf{B}^*)}(v))\cdot\sum_{\mathbf{H}\setminus\mathbf{H_D}}\prod_{v\in(\mathbf{H}\setminus\mathbf{H_D})}p(v|\,\mathrm{pa}_{\mathcal{G}(\mathbf{O}\cup\mathbf{H},\mathbf{B}^*)}(v))\\
&=\sum_\mathbf{H}\prod_{v\in\mathbf{D}\cup\mathbf{H_D}}p(v|\,\mathrm{pa}_{\mathcal{G}(\mathbf{O}\cup\mathbf{H},\mathbf{B}^*)}(v))\\
&=\sum_\mathbf{H}\phi_{\mathbf{D}^*\setminus\mathbf{D}}(q(\mathbf{D}^*|\,\mathrm{pa}_{\mathcal{G}(\mathbf{O}\cup\mathbf{H},\mathbf{B}^*)}(\mathbf{D}^*)));\mathcal{G}(\mathbf{O}\cup\mathbf{H},\mathbf{B}^*))
\end{aligned}\tag{3}
$$

Once again, these equalities are a result of the above constructions of $\mathbf{H}$ and $\mathbf{H}^*$. By commutativity (Lemma 55 in [2]), we can remove references to $\mathbf{H}$:

$$
\begin{aligned}
p(\mathbf{Y}^*&|do_{\mathcal{G}(\mathbf{O}\cup\mathbf{H},\mathbf{B}^*)}(\mathbf{A}^*))\\
&=\prod_{\mathbf{D}\in\mathcal{D}(\mathcal{G}(\mathbf{Y}^*,\mathbf{B}^*\cup\mathbf{A}^*))}\phi_{\mathbf{D}^*\setminus\mathbf{D}}q(\mathbf{D}^*|\,\mathrm{pa}_{\mathcal{G}(\mathbf{O},\mathbf{B}^*)}(\mathbf{D}^*));\mathcal{G}(\mathbf{O},\mathbf{B}^*))\\
&=\prod_{\mathbf{D}\in\mathcal{D}(\mathcal{G}(\mathbf{Y}^*,\mathbf{B}^*\cup\mathbf{A}^*))}\phi_{\mathbf{D}^*\setminus\mathbf{D}}q(\mathbf{D}^*|\,\mathrm{pa}_{\mathcal{G}}(\mathbf{D}^*));\mathcal{G}^d)\\
&=\prod_{\mathbf{D}\in\mathcal{D}(\mathcal{G}_{\mathbf{Y}^*})}\phi_{\mathbf{D}^*\setminus\mathbf{D}}q(\mathbf{D}^*|\,\mathrm{pa}_{\mathcal{G}}(\mathbf{D}^*));\mathcal{G}^d)
\end{aligned}
$$

The second equality is true because $\mathrm{pa}_{\mathcal{G}}(\mathbf{D}^*)\subseteq\mathrm{pa}_{\mathcal{G}(\mathbf{O},\mathbf{B}^*)}(\mathbf{D}^*)$ and by the assumption of a block-safe chain graph. The final equality is true by block-safeness and the definition of induced subgraphs.

Finally by the fact that $p(\mathbf{Y}|do_{\mathcal{G}(\mathbf{O}\cup\mathbf{H},\mathbf{B}^*)}(\mathbf{A})) = \sum_{\mathbf{Y}^*\setminus\mathbf{Y}}p(\mathbf{Y}^*|do_{\mathcal{G}(\mathbf{O}\cup\mathbf{H},\mathbf{B}^*)}(\mathbf{A}^*))$, we can rewrite the above as:

$$
p(\mathbf{Y}|do_{\mathcal{G}(\mathbf{O}\cup\mathbf{H},\mathbf{B}^*)}(\mathbf{A})) = \sum_{\mathbf{Y}^*\setminus\mathbf{Y}}\prod_{\mathbf{D}\in\mathcal{D}(\mathcal{G}_{\mathbf{Y}^*})}\phi_{\mathbf{D}^*\setminus\mathbf{D}}q(\mathbf{D}^*|\,\mathrm{pa}_{\mathcal{G}}(\mathbf{D}^*));\mathcal{G}^d)
$$

We combine this with the block portioned derived above via chain-graph g-formula to obtain the result of the sub-claim

**Claim 3**: *If there is a district in $\mathcal{D}(\mathcal{G}(\mathbf{O})_{\mathbf{Y}^*})$ that is not reachable in $\mathcal{G}^d$, then $p(\mathbf{Y}|do(\mathbf{a}))$ is not identifiable.*

Let $\mathbf{D} \in \mathcal{D}(\mathcal{G}(\mathbf{O})_{\mathbf{Y}^*})$ be unreachable. Let $\mathbf{R} = \{D \in \mathbf{D} | \operatorname{ch}_{\mathcal{G}}(D) \cap \mathbf{D} = \emptyset\}$. Let $\mathbf{A}^* = A \cap \operatorname{pa}_{\mathcal{G}}(D)$. Then there exists a superset of $\mathbf{D}$, $\mathbf{D}'$, such that $\mathbf{D}$ and $\mathbf{D}'$ form a hedge for $p(\mathbf{R}|do(\mathbf{a}^*))$ and thus $p(\mathbf{R}|do(\mathbf{a}^*))$ is not identified [3].

Let $\mathbf{Y}'$ be the minimal subset of $\mathbf{Y}$ such that $\mathbf{R} \subseteq \operatorname{ant}_{\mathcal{G}(\mathbf{O})_{\mathbf{O}\setminus\mathbf{A}}}(\mathbf{Y}')$. Consider an edge subgraph $\mathcal{G}^\dagger$ of $\mathcal{G}$ consisting of all edges in $\mathcal{G}$ in the hedge formed by $\mathbf{D}, \mathbf{D}'$ and edges on partially directed paths in $\mathcal{G}(\mathbf{O})_{\mathbf{O}\setminus\mathbf{A}}$ from every element in $\mathbf{R}$ to some element in $\mathbf{Y}'$, such that the edge subgraph does not contain any cycles (directed or otherwise).

We proceed as follows. We first define an ADMG $\tilde{\mathcal{G}}^\dagger$ from $\mathcal{G}^\dagger$ as follows. The vertices and edges making up the hedge structure [3] in $\mathcal{G}^\dagger$ are also present in $\tilde{\mathcal{G}}^\dagger$. For every partially directed path $\sigma$ from an element in $\mathbf{R}$ to an element in $\mathbf{Y}'$, we construct a directed path from $\mathbf{R}$ in $\tilde{\mathcal{G}}^\dagger$ containing vertex copies of vertices on the undirected path $\sigma$, and which orients all undirected edges in $\sigma$ away from $\mathbf{R}$ and towards the element copy in $\tilde{\mathcal{G}}^\dagger$ of the appropriate element of $\mathbf{Y}'$ in $\mathcal{G}^\dagger$.

We then prove non-identifiability of $p(\tilde{\mathbf{Y}}'|do(\mathbf{a}^*))$ in $\tilde{\mathcal{G}}^\dagger$, where $\tilde{\mathbf{Y}}'$ is the set of all vertex copies in $\tilde{\mathcal{G}}^\dagger$ of vertices in $\mathbf{Y}'$ in $\mathcal{G}^\dagger$, using standard techniques for ADMGs. In particular, we follow the proof of Theorem 4 in the supplement of [4].

We next show that $p(\mathbf{Y}' \mid do(\mathbf{a}^*))$ is not identified in $\mathcal{G}^\dagger$. For the two counterexamples in the causal model given by $\tilde{\mathcal{G}}^\dagger$ witnessing non-identifiability of $p(\tilde{\mathbf{Y}}' \mid do(\mathbf{a}^*))$ in the above proof, we will construct two counterexamples in the causal model given by $\mathcal{G}^\dagger$ witnessing non-identifiability of $p(\mathbf{Y}' \mid do(\mathbf{a}^*))$.

To do so, we define new variables along all partially directed paths from $\mathbf{R}$ to $\mathbf{Y}'$ in $\mathcal{G}^\dagger$ as Cartesian products of variable copies in counterexamples constructed. Note that any such variable containing only a single element in $\mathbf{R}$ in its anterior in $\mathcal{G}^\dagger$ will only have a single copy, while a variable containing two elements in $\mathbf{R}$ in its anterior in $\mathcal{G}^\dagger$ will contain two copies, and so on. It's clear that the two resulting elements contain vertices in $\mathcal{G}^\dagger$, agree on the observed data distribution, and disagree on $p(\mathbf{Y}' \mid do(\mathbf{a}^*))$.

What remains to show is that the distributions so constructed obey one of CG Markov properties associated with a CG $\mathcal{G}^\dagger$. Fix a (possibly trivial) block $\mathbf{B}$ in $\mathcal{G}^\dagger$. We must show for each $B \in \mathbf{B}$ that $p(B \mid \mathbf{B} \setminus B, \operatorname{pa}_{\mathcal{G}^\dagger}(\mathbf{B})) = p(B \mid \operatorname{nb}_{\mathcal{G}^\dagger}, \operatorname{pa}_{\mathcal{G}}(B))$.

For any $B \in \mathbf{B}$ in $\mathcal{G}^\dagger$, there exists a set $B_1, \ldots, B_k$ of variables in $\tilde{\mathcal{G}}^\dagger$ such that $B$ is defined as $B_1 \times \ldots \times B_k$. Moreover, any variable $A \in \operatorname{nb}_{\mathcal{G}^\dagger}(B) \cup \operatorname{pa}_{\mathcal{G}^\dagger}(B)$ corresponds to a Cartesian product $A_1 \times A_m$ of variables where $A_i$ is a child or a parent of some variables $B_j$. The result then follows by d-separation in $\tilde{\mathcal{G}}^\dagger$, and the fact that the part of $\tilde{\mathcal{G}}^\dagger$ outside of the hedge structure does not contain any colliders by construction. □

## 2 Derivations

Consider Figure 1 (a). We are interested in identifying $p(Y_2(a_1, a_2))$. We set $\mathbf{Y}^*$ to the anterior of $\mathbf{Y}$ in $\mathcal{G}_{\mathbf{V}\setminus\mathbf{A}}$: $\mathbf{Y}^* \equiv \{C_1, C_2, M_1, M_2, Y_2\}$ (see $\mathcal{G}_{\mathbf{Y}^*}$ shown in Fig. 1 (b)) with $\mathcal{B}(\mathcal{G}_{\mathbf{Y}^*}) = \{\{M_1, M_2\}\}$ and $\mathcal{D}(\mathcal{G}_{\mathbf{Y}^*} = \{\{C_1\}, \{C_2\}, \{Y_2\}\}$. We can now proceed with the version of the ID algorithm for SGs. The CCG portion of the algorithm simply yields $p(M_1, M_2|A_1 = a_1, A_2, C_1, C_2)$. Note that this expression further factorizes according to the factorization of blocks in a chain graph. For the ADMG portion of the algorithm, we must fix variables in three different sets $\{C_2, A_1, A_2, Y_1, Y_2\}$, $\{C_1, A_1, A_2, Y_1, Y_2\}$, $\{C_1, C_2, A_1, A_2, Y_1\}$ in $\mathcal{G}^d$, shown in Fig. 1 (c), corresponding to three

Figure 1: (a) A latent projection of the CG in (Fig. 1a in the main paper) onto observed variables. (b) The graph representing $\mathcal{G}_{\mathbf{Y}^*}$ for the intervention operation $do(a_1)$ applied to (a). (c) The ADMG obtained by fixing $M_1, M_2$ in (a).

districts in Fig. 1 (b). We have:

$$
\begin{aligned}
& \phi_{\{C_2,A_1,A_2,Y_1,Y_2\}}(p(Y_1,Y_2|A_1,A_2,M_1,M_2,C_1,C_2)p(A_1,A_2,C_1,C_2)) \\
& = \phi_{\{C_2,A_1,A_2,Y_1\}}(p(Y_1|A_1,A_2,M_1,M_2,C_1,C_2,Y_2)p(A_1,A_2,C_1,C_2)) \\
& = \phi_{\{C_2,A_1,A_2\}}(p(A_1,A_2,C_1,C_2)) \\
& = \phi_{\{C_2,A_2\}}(p(A_2,C_1,C_2)) \\
& = \phi_{\{C_2\}}(p(C_1,C_2)) \\
& = p(C_1)
\end{aligned}
\tag{4}
$$

$$
\begin{aligned}
& \phi_{\{C_1,A_1,A_2,Y_1,Y_2\}}(p(Y_1,Y_2|A_1,A_2,M_1,M_2,C_1,C_2)p(A_1,A_2,C_1,C_2)) \\
& = \phi_{\{C_1,A_1,A_2,Y_1\}}(p(Y_1|A_1,A_2,M_1,M_2,C_1,C_1,Y_2)p(A_1,A_2,C_1,C_2)) \\
& = \phi_{\{C_1,A_1,A_2\}}(p(A_1,A_2,C_1,C_2)) \\
& = \phi_{\{C_1,A_2\}}(p(A_2,C_1,C_2)) \\
& = \phi_{\{C_1\}}(p(C_1,C_2)) \\
& = p(C_2)
\end{aligned}
\tag{5}
$$

$$
\begin{aligned}
& \phi_{\{C_1,C_2,A_1,A_2,Y_1\}}(p(Y_1,Y_2|A_1,A_2,M_1,M_2,C_1,C_2)p(A_1,A_2,C_1,C_2)) \\
& = \phi_{\{A_1,Y_1,A_2\}}(p(Y_1,Y_2|A_1,A_2,M_1,M_2,C_1,C_2)p(A_1,A_2|C_1,C_2)) \\
& = \phi_{\{A_1,A_2\}}(p(Y_2|A_1,A_2,M_1,M_2,C_1,C_2)p(A_1,A_2|C_1,C_2)) \\
& = \sum_{A_2} p(Y_2|A_1,A_2,M_1,M_2,C_1,C_2)p(A_2|C_2) \\
& = \sum_{A_2} p(Y_2|A_1,A_2,M_2,C_2)p(A_2|C_2)
\end{aligned}
\tag{6}
$$

with the last term evaluated at $A_1 = a_1$. Thus, the identifying functional is:

$$
p(Y_2(a_1,a_2)) = \sum_{\{C_1,C_2,M_1,M_2\}} \Bigg[ p(M_1,M_2|a_1,a_2,C_1,C_2) \\
\times \Bigg[ \sum_{A_2} p(Y_2|a_1,A_2,M_2,C_2)p(A_2|C_2)p(C_1)p(C_2) \Bigg] \Bigg]
\tag{7}
$$

## 3 Simulation Study

### 3.1 The Auto-G-Computation Algorithm

To estimate identifying functionals corresponding to causal effects given dependent data, we generally use maximum likelihood plug in estimation. The exception is the factor $p(\mathbf{M} \mid \mathrm{pa}_{\mathcal{G}}(\mathbf{M}))$, which may not be estimated if $M_i$ variables for all units $i$ are dependent, as is the case in our simulation study. In this case, the above density must be estimated from a single sample. Thus, standard statistical methods such as maximum likelihood estimation fail to work. We adapt the auto-g-computation algorithm method in [5], which exploits Markov assumptions embedded in our CG model, as well as the pseudo-likelihood or coding estimation methods introduced in [1]. We briefly describe the approach here.

The auto-g-computation algorithm is a generalization of the Monte Carlo sampling version of the standard g-computation algorithm for classical causal models (represented by DAGs) [6] to causal models represented by CGs. Auto-g-computation proceeds by generating samples from a block using Gibbs sampling. The parameters for Gibbs factors used in the sampler (which, by the global Markov property for CGs, take the form of $p(X_i \mid \mathrm{pa}_{\mathcal{G}}(X_i) \cup \mathrm{nb}_{\mathcal{G}}(X_i))$) are learned via parameter sharing and coding or pseudo-likelihood based estimators. For any block $\mathbf{B}$, the Gibbs sampler draws samples from $p(\mathbf{X} \mid \mathrm{pa}_{\mathcal{G}}(\mathbf{X}))$, given a fixed set of samples drawn from all blocks with elements in $\mathrm{pa}_{\mathcal{G}}(\mathbf{X})$, or specific values of $\mathrm{pa}_{\mathcal{G}}(\mathbf{X})$ we are interested in, as follows.
Gibbs Sampler for $\mathbf{X}$:

$$\text{for } t = 0, \text{let } \mathbf{x}^{(0)} \text{ denote initial values ;}$$
$$\text{for } t = 1, ..., T$$
$$\text{draw value of } X_1^{(t)} \text{ from } p(X_1 | \mathbf{x}_{\mathrm{pa}_{\mathcal{G}}(X_1) \cup \mathrm{nb}_{\mathcal{G}}(X_1)}^{(t-1)}));$$
$$\text{draw value of } X_2^{(t)} \text{ from } p(X_2 | \mathbf{x}_{\mathrm{pa}_{\mathcal{G}}(X_2) \cup \mathrm{nb}_{\mathcal{G}}(X_2)}^{(t-1)}));$$
$$\vdots$$
$$\text{draw value of } X_m^{(t)} \text{ from } p(X_m | \mathbf{x}_{\mathrm{pa}_{\mathcal{G}}(X_m) \cup \mathrm{nb}_{\mathcal{G}}(X_m)}^{(t-1)}));$$

Since we are interested in estimating a functional similar to (7), we use observed values of $\mathbf{C}$, and intervened on values $a_i, a_j$ as the values of $\mathrm{pa}_{\mathcal{G}}(\mathbf{M})$ in the Gibbs sampler.

The coding-likelihood and pseudo-likelihood estimators we use are described in more detail in [5]. Both estimators rely on parameter sharing for densities $p(M_i \mid \mathrm{pa}_{\mathcal{G}}(M_i) \cup \mathrm{nb}_{\mathcal{G}}(M_i))$ across different units $i$, and for the network to be sufficiently sparse such that each $M_i$ depends on only a few other variables in the model, relative to the total number of units.

The coding estimator uses a subset of the data that corresponds to units that form independent sets in the network adjacency graph (where units are adjacent of they are friends in the network, and not adjacent otherwise). A set of units is a *maximal* independent set in the network adjacency graph if a) no two vertices in the set are adjacent, and b) it is impossible to add another unit to the set without violating the adjacency constraint. A *maximum* independent set is a maximal independent set such that there does not exist a larger maximal independent set in the same graph. Finding maximum independent sets is a classic NP-complete problem; in practice we find several *maximal* independent sets and pick the one with largest cardinality as a heuristic. See Table 1 below for the size of $S_{max}$ for each network size in our experiments. The coding likelihood estimator was proven consistent

| $N$ | 400 | 800 | 1000 | 2000 |
|---|---|---|---|---|
| $|S_{max}|$ | 159 | 309 | 384 | 763 |

Table 1: The size of $S_{max}$ used for the coding-likelihood estimator in each network

and asymptotically normal in [5] whereas pseudo-likelihood estimation is, under mild assumptions, consistent but not asymptotically normal. On the other hand, pseudo-likelihood estimation is more efficient than coding likelihood estimation since it makes use of all of the data.

## 3.2 Simulation Specifics

For data generation we use the following densities for $A_i, M_i, Y_i$, parameterized by $\tau_A = \{\gamma_0, \gamma_{C_1}, \ldots, \gamma_{C_p}, \gamma_{U_1}, \ldots, \gamma_{U_q}\}, \tau_M = \{\beta_0, \beta_A, \beta_{C_1}, \ldots, \beta_{C_p}\beta_{A_{nb}}, \beta_{M_{nb}}\}, \tau_Y = \{\alpha_0, \alpha_{C_1}, \ldots, \alpha_{C_p}, \alpha_{U_1}, \ldots, \alpha_{U_q}, \alpha_{A_{nb}}, \alpha_M\}$:

$$p(A_i = 1|\mathbf{C}_i, \mathbf{U}_i; \tau_A) = expit(\gamma_0 + \big(\sum_{l=1}^{p} \gamma_{C_l} C_{il}\big) + \big(\sum_{l=1}^{q} \gamma_{U_l} U_{il}\big))$$

$$p(M_i = 1|A_i, \mathbf{C}_i, \{A_j, M_j | j \in \mathcal{N}_i\}; \tau_M)$$
$$= expit(\beta_0 + \beta_A A_i + \big(\sum_{l=1}^{p} \beta_{C_l} C_{il}\big) + \big(\sum_{j \in \mathcal{N}_i} (\beta_{A_{nb}} A_j + \beta_{M_{nb}} M_j)\big))$$

$$p(Y_i = 1|\mathbf{C}_i, \mathbf{U}_i, M_i, \{A_j | j \in \mathcal{N}_j\}; \tau_Y)$$
$$= expit(\alpha_0 + \big(\sum_{l=1}^{p} \alpha_{C_l} C_{il}\big) + \big(\sum_{l=1}^{q} \alpha_{U_l} U_{il}\big) + \big(\sum_{j=\mathcal{N}_i} \alpha_{A_{nb}} A_j\big) + \alpha_M M_i).$$

The values of the parameters for the beta distributions we use to generate $\mathbf{C}_i, \mathbf{U}_i$ can be found in Table 2a while the values of $\tau_A, \tau_M, \tau_Y$ can be found in Table 2b.

| Variable | a | b |
|---|---|---|
| $C_1$ | 1.5 | 3 |
| $C_2$ | 6 | 2 |
| $C_3$ | 0.8 | 0.8 |
| $U_1$ | 2.3 | 1.1 |
| $U_2$ | 0.9 | 1.1 |
| $U_3$ | 2 | 2 |

(a) Parameters for $\mathbf{C}$ and $\mathbf{U}$

| Parameter | Value |
|---|---|
| $\tau_A$ | (-1, 0.5, 0.2, 0.25, 0.3, -0.2, 0.25) |
| $\tau_M$ | (-1, -0.3, 0.4, 0.1, 1, -0.5, -1.5) |
| $\tau_Y$ | (-0.3, -0.2, 0.2, -0.05, 0.1, -0.2, 0.25, -1, 3) |

(b) Parameters for $\tau_A, \tau_M, \tau Y$

Table 2: The parameters for each generating distribution

## 3.3 Extended Results

In the main paper we gave confidence intervals and the mean and standard deviation of the bias of our estimators. All results were calculated by averaging over 1000 simulated networks.

| Ground Truth Network Average Effects | | | | |
|---|---|---|---|---|
| $N$ | 400 | 800 | 1000 | 2000 |
| Ground Truth | -.455 | -.453 | -.455 | -.456 |

Table 3: The ground truth effects for each network, calculated by averaging over 5 samples of the data generating process for each network under the relevant interventions

As discussed in the main body of the paper, the estimators we use are able to recover the effects of interest reasonably well. The approximate ground truth values for these effects can be found in Table 3. The fact that the coding estimator restricts the network to a small fraction of its total units means it is considerably less efficient than the pseudo-likelihood estimator.

Though the pseudo-likelihood estimator is not in general asymptotically normal, it does not perform substantially worse than the provably asymptotically normal coding-likelihood estimator. In both cases, the true effect is covered by the 95% confidence interval of the estimator.