[Reviews · NeurIPS 2018]

Reviewer 1



The authors develop theory for causal inference in non-iid setting where samples are divided to blocks, based on segregated graphs (no partially directed cycles, no vertex with both neighbours and siblings). The paper includes extensive theory and a real data application. The paper is written very clearly. This is interesting research, studying causal effect identification in more general settings is likely to have impact. Why would data dependence be symmetric and not asymmetric? – This would need further justification. Also different chain graph interpretations correspond to different types of symmetric relationships, does that make a difference here? Graph theory: meaning of a sibling and neighbor should be defined. Figure 1 caption leaves f and g undescribed. Also https://link.springer.com/article/10.1007/s41237-018-0051-2 discusses causal effect identification in a type of CGs. This paper discusses causality in graph with similar undirected, bidirected and directed edges http://proceedings.mlr.press/v52/pena16.pdf. I think these works should be cited. p.2 I believe undirected and bidirected path are undefined and can be interpreted in several ways(depending on the direction intermediate edges): “path of bidirected edges”, “path of undirected edged” may be better?

Reviewer 2



The work presented in the paper is clearly of value. The existing theory for identification and estimation of causal parameters in DAGs for IID data has been central to our understanding of causal inference, and developing analogous results for data under interference would be useful both to apply directly to data in which we know interference occurs and to better understand the potential impacts of violations of the IID assumption. While the paper should be accepted, the current version could be substantially improved in both organization and in its discussion of several key issues, including generality, assumptions, temporal effects, and prior work. Organization and Presentation Some aspects of the organization make the paper challenging for readers. Some sections do not provide a “roadmap” to the basic logic before plunging into the details, others do not present a high-level intuition for why a given theoretical result is being presented, The entire paper would be substantially improved if the authors provided readers with a high-level roadmap to the overall reasoning of the paper, making clear the basic logic that allows an identification theory to be developed under interference (before plunging into the details of sections 2 and 3). As I understand it, the outline of that logic is: 1) Represent models as latent variable chain graphs, 2) Divide the model into blocks, 3) Assume no interference between blocks, 4) Express identification theory by using truncated nested factorization of latent projection ADMGs. This logic differs from the current abstract because it helps readers see how the paper achieves what it does, rather than just what the paper achieves. The paper provides a great deal of theory with relatively little intuition to help readers understand the underlying concepts. Such intuition would be relatively easy to provide. For example, the fixing operation will be novel to most readers, and the paper would be improved by providing an intuition for what fixing corresponds to (if such an intution exists). The intuition for conditioning, for example, is often presented in terms of partitioning of possible worlds. If no such intuition exists, then it would be worth explaining up front that it is only a method for producing a nested factorization of an ADMG (or even relegating this to the supplemental material). The abtract implies that unobserved confounding bias is a necessary (or at least likely) consequence of non-iid data, but the paper doesn’t make clear why this is so. A quick description would be useful, so that readers would not have to consult outside papers, such as reference 14. In contrast, Section 1 merely invokes latent confounding as a likely situation that appears in conjunction with interference. More clarity on this point would help readers. Generality It is very difficult to assess the generality of the results presented in the paper. Specifically, the details of the motivating example provided in Section 2 makes the results appear very narrow, but the more expansive langauge in the introduction and conclusion make the results seem much more general (“…complete identification theory, and estimation for dependent data.”). Specifically, the work presented here applies to data consisting of a single object type and stable, symmetric relationships, such as friendship ties. This is worth emphasizing in the introduction, because interference can occur in cases in which asymmetric relationships also occur (such as social network data, communications data, and citation data) and where multiple types of objects occur. Assumptions The method outlined in the paper appears to rely strongly on the presence of blocks (maximal sets of vertices in the model which each pair is connected by an undirected path) that lack interference. This seems unlikely in many reasonable scenarios. Interblock non-interference may be a common assumption, but that doesn’t mean that it is realistic. The authors should either provide evidence that this is a plausible assumption in real applications or provide evidence that moderate violations of this assumption do not greatly affect the results. Section 2 states that “a crucial assumption in our example is that…community membership does not directly influence sharing, except as mediated by social network activity of the user.” This seems a moderately strong assumption, and it would be good to expand somewhat on both the likelihood of being able to gather data that meet such an assumption and the reasons why this assumption is made. Temporal effects The motivating example (and other plausible examples) seems extremely difficult to analyze without explicit consideration of the time-series of behavior. This is not just a simple statement that temporal information aids causal identification. Instead, the point here is that the system described almost certainly has substantial feedback effects and time-varying exogenous influences, and it may never reach any form of equilibrium. Thus, it is unclear what a data sample corresponds to here. The paper would be improved greatly by describing in more detail the underlying assumptions of data collection. Prior work The paper ignores a moderately large literature on inferring causal models of non-iid data that has appeared in the graphical models literature over the past decade. Relevant papers include: Marc Maier (2013). Reasoning about independence in probabilistic models of relational data. arXiv:1302.4381. Lee & Honavar (2016). On learning causal models from relational data. AAAI 2016. Maier et al. (2013). A sound and complete algorithm for learning causal models from relational data. UAI 2013. Arbour et al. (2016). Inferring network effects from observational data. KDD 2016. These papers do not subsume or contradict the current paper, but the current paper would be improved by contrasting the assumptions of these papers with the approach taken here. Summarize the strengths and weaknesses of the submission, focusing on each of the following four criteria: quality, clarity, originality, and significance.

Reviewer 3



Thanks for the rebuttal from the authors. I have read the rebuttal. I have adjusted my rating accordingly. =============== This paper addresses the challenge of causal inference under both interference and unmeasured confounding. A classical solution to interference is chain graph; a framework for unmeasured confounding is latent variable DAG. Hence, this work tries to bring together the two to develop latent variable chain graphs. It develops the identification conditions, provides estimation algorithm, and performs experiments. The paper is nicely written and well motivated. The network example in the introduction is illuminating. The technical content of the paper is also self-contained, especially for reading unfamiliar with segregated projection. One place that could be elaborated is in ADMG. the condition for a double arrow to exist is rather abstract (and a bit hard to parse). Maybe more explanation on Fig. 1 could help. However, the empirical studies is not convincing yet. The study contains one simulation of different network sizes. The estimates do not seem close to the ground truth values. the confidence intervals also do not contain the truth values. While the paper makes conceptual contribution, it is not yet clear how well the proposed method solve the problem. Lastly, a few question of interest remains open here: 1. Does unmeasured confounding interact with network interference? 2. Does dependent data exhibit other complications than unmeasured confounding and interference? 3. Chain graphs are known to be computational expensive. Does unmeasured confounding worsen the situation? How expensive is the computation in the simulation studies? 4. This work requires community membership does not directly influence sharing, except as mediated by social network activity of the user. How restrictive is this condition? (Without this assumption, does it imply some form of interaction between unmeasured confounding and interference?) 5. How does dependent data move beyond interference? Does the dependence exacerbate the unmeasured confounding issue? How?